# Interfering with the ERC1–LL5β interaction disrupts plasma membrane–Associated platforms and affects tumor cell motility

**Lucrezia Maria Ribolla**[1,☯], **Kristyna Sala**[1,☯], **Diletta Tonoli**[1], **Martina Ramella**[1], **Lorenzo Bracaglia**[2], **Isabelle Bonomo**[3], **Leonardo Gonnelli**[2], **Andrea Lamarca**[1], **Matteo Brindisi**[1], **Roberta Pierattelli**[2], **Alessandro Provenzani**[3], **Ivan de Curtis**[1]*

**1** Vita-Salute San Raffaele University and San Raffaele Scientific Institute, Milano, Italy, **2** Department of Chemistry "Ugo Schiff" and Magnetic Resonance Center, University of Florence, Sesto Fiorentino (Florence), Italy, **3** Department of Cellular, Computational and Integrative Biology, University of Trento, Trento, Italy

☯ These authors contributed equally to this work.
* decurtis.ivan@hsr.it

**Data Availability Statement:** The set of data necessary to replicate the study findings are available in the San Raffaele Open Research Data

## Abstract

Cell migration requires a complex array of molecular events to promote protrusion at the front of motile cells. The scaffold protein LL5β interacts with the scaffold ERC1, and recruits it at plasma membrane–associated platforms that form at the front of migrating tumor cells. LL5 and ERC1 proteins support protrusion during migration as shown by the finding that depletion of either endogenous protein impairs tumor cell motility and invasion. In this study we have tested the hypothesis that interfering with the interaction between LL5β and ERC1 may be used to interfere with the function of the endogenous proteins to inhibit tumor cell migration. For this, we identified ERC1(270–370) and LL5β(381–510) as minimal fragments required for the direct interaction between the two proteins. The biochemical characterization demonstrated that the specific regions of the two proteins, including predicted intrinsically disordered regions, are implicated in a reversible, high affinity direct heterotypic interaction. NMR spectroscopy further confirmed the disordered nature of the two fragments and also support the occurrence of interaction between them. We tested if the LL5β protein fragment interferes with the formation of the complex between the two full-length proteins. Coimmunoprecipitation experiments showed that LL5β(381–510) hampers the formation of the complex in cells. Moreover, expression of either fragment is able to specifically delocalize endogenous ERC1 from the edge of migrating MDA-MB-231 tumor cells. Coimmunoprecipitation experiments show that the ERC1-binding fragment of LL5β interacts with endogenous ERC1 and interferes with the binding of endogenous ERC1 to full length LL5β. Expression of LL5β(381–510) affects tumor cell motility with a reduction in the density of invadopodia and inhibits transwell invasion. These results provide a proof of principle that interfering with heterotypic intermolecular interactions between components of plasma membrane–associated platforms forming at the front of tumor cells may represent a new approach to inhibit cell invasion.

Repository (ORDR, https://ordr.hsr.it/research-data/) with the DOI 10.17632/st7ggs4s92.1.

**Funding:** IdC was supported by Fondazione AIRC per la Ricerca sul Cancro (IG 2017 Id.20203), and by Instruct-ERIC (PID: 17416). KS and MR were supported by postdoctoral fellowships from Fondazione AIRC per la Ricerca sul Cancro (Fellowships 22253 and 24231, respectively). LMR was supported by a FCSR-Fronzaroli fellowship; work performed by LMR was in partial fulfilment of the requirements for obtaining the PhD degree at the Vita-Salute San Raffaele University, Milano, Italy. The funders had no role in study design, data collection and analysis, decision to publish, or preparation of the manuscript.

**Competing interests:** The authors have declared that no competing interests exist.

## Introduction

Plasma membrane-associated platforms (PMAPs) are assembled on the cytoplasmic side of specific sites of the cell membrane by core proteins including the scaffold and adaptors Liprin-α1, ERC1/ELKS and LL5β [1]. Liprin-α and ERC/ELKS proteins are established players essential for the assembly of functional presynaptic sites in neurons [2], while LL5α and LL5β, also known as pleckstrin homology-like domain family B member 1 (PHLDB1) and 2 (PHLDB2) respectively, are ubiquitous scaffold proteins with a carboxy-terminal pleckstrin homology (PH) domain specific for binding to phosphatidylinositol-3,4,5-trisphosphate (PIP3) at the leading edge of migrating cells, where it promotes actin polymerization required for protrusion [3, 4]. Mammalian LL5α and LL5β share an overall 38% identity, with highest conservation in the carboxy-terminal PH domain. LL5α has an amino-terminal Forkhead-associated (FHA) domain, a phospho-threonine binding module found in several proteins. A number studies suggest that LL5α and LL5β play redundant roles. For example, they are both involved in microtubule anchoring to laminin-based cell adhesions [5], in maintaining basal membrane integrity in lateral epiblast cells [6], and in supporting tumor cell migration and invasion [7]. On the other hand, LL5α was found to play a specific role during adipocyte differentiation: this protein mediates the insulin stimulation of Akt kinase phosphorylation, and silencing of LL5α, but not of LL5β, attenuates insulin-stimulated deoxyglucose transport and glucose transporter GLUT4 translocation to the plasma membrane [8].

LL5β is a filamin binding protein and an effector of the phosphoinositide 3-kinase (PI3K) signaling pathway [9]. At high, PDGF-stimulated PIP3 levels, LL5β localizes at the plasma membrane, while after serum starvation or incubation with the PI3K inhibitor wortmannin the protein is redistributed to cytoplasmic structures defined as vesicles, although their vesicular nature has never been proved [10]. LL5β appears thus to act as a sensor of levels of PIP3. Human LL5β is made of 1253 amino acid residues, with a 33-amino acid region (residues 323–356) required for binding to filamin [11] and localization to stress fibers and cortical F-actin [9].

The region including residues 306–562 of LL5β interacts directly with the region including residues 200–400 of ERC1/ELKS, while residues 563–916 of LL5β interact directly with CLASP [12]. ERC1 is a coiled-coil rich scaffold protein with a predicted amino-terminal intrinsically disordered region (IDR) [13]. ERC1 and LL5 proteins are part of protein complexes including the adaptor protein Liprin-α1 [14, 15]. In non-motile cells this complex has been shown to anchor microtubules to the cell cortex [12]. Thus, LL5β may link and coordinate the actin and microtubule cytoskeletons by interacting on one side with the actin-binding protein filamin, and on the other side with the cytoplasmic linker-associated proteins (CLASPs) that bind to the plus ends of microtubules to control their dynamics [16].

The localization of LL5 proteins at protrusions in migrating tumor cells favors the assembly of PMAPs by interacting with other PMAP components [7]. Of note, PMAPs form both near lamellipodia, the flat actin-rich structures at the very edge of migrating cells required for protrusion [7], and near invadosomes (including invadopodia and podosomes) [17], the protruding structures secreting metalloproteases that are needed to digest the extracellular matrix and allow tumor cell migration and invasion [18].

In this study, we have addressed the interaction between the PMAP proteins LL5β and ERC1, and identified specific fragments required for this interaction. Biochemical characterization and NMR experiments confirm the intermolecular interaction and indicate a high affinity binding between the two proteins. Expression of either fragment in invasive human breast cancer cells perturbs the formation of PMAPs at the edge of migrating cells. Moreover, the ERC1-binding fragment LL5β(381–510) affects tumor cell motility with a reduction in the

density of invadopodia and inhibition of invasion *in vitro*. Our results provide a proof of principle that interfering with intermolecular interactions involved in PMAPs assembly may be used to perturb tumor cell invasion.

## Materials and methods

### Plasmids

Plasmids for GFP-ERC1 (murine ELKSε, 1116 residues) [15], GFP-LL5β and Cherry-LL5β (murine isoform 3, 1206 residues) [19], FLAG-Liprin-α1 (human), FLAG-Liprin-ΔEBR, FLAG-Liprin-EBR, GFP-LL5β-PHm, GFP-ERC1-N, His-ERC1-N, GFP-ERC1-C, His-ERC1-C, GFP-ERC1Δ147 and FLAG-βGal were described previously [7, 13, 20–22]. ERC1 fragments ERC1(199–402), ERC1(199–270), ERC1(270–370), ERC1(199–335), ERC1(316–402), and ERC1(262–402) were obtained by PCR from the plasmid pGFP-C1-ERC1, and subcloned into the pGFP-C1 or mCherry-C1 vector (Clontech Laboratories). LL5β fragments LL5β(305–558), LL5β(305–450), LL5β(381–510), LL5β(448–558), LL5β(381–558), LL5β(381–470), and LL5β (425–558) were obtained by PCR from the plasmid pmCHERRY-LL5b and subcloned into the pFLAG-CMV2 vector (Kodak). For PCR, the following primers were used: GFP-ERC1(199–402): forward 5′-GGAATTCCATGG CCTTGAGAAAAGATGAAGC-3′, reverse 5′-GGAATT CTTATTCCAGGTCTCGGAGC-3′; GFP-ERC1(199–270): forward 5′-GGAATTCCATGGCC TTGAGAAAAGATGAAGC-3′, reverse 5′-GGAATTCTTACAGCCTCTGGAAGTTCTCC-3′; GFP-ERC1(270–370): forward 5′-GGAATTCC ATGCTGCATGCTGAGCACG-3′, reverse 5′-GGAATTCTTAATTTTCAAATCTCCGGTGC-3′; GFP-ERC1(199–335): forward 5′-GGA ATTCCATGGCCTTGAGAAAAGATGAAGC-3′, reverse 5′-GGAATTCTTACCGTCGCGTT CTCTCATG-3′; GFP-ERC1(316–402): forward 5′-GGAATT CCATGCTCCAGAGCAAAG GAC-3′, reverse 5′-GGAATTCTTATTCCAGGTCTCGGAGC-3′; GFP-ERC1(262–402): forward 5′-GGAATTCCATGCTCACAGAGGAGAACTTCCAG-3′; reverse 5′- GGAATTCTTA TTCCAGGTCTCGGAGC-3′; FLAG-LL5β(305–558): forward 5′-GGAATTCCATGTCTCTAA GCTCAGGGGC-3′; reverse 5′-GGAATTCTTAGGTTGCTTTCAGAA ACGC-3′; FLAG-LL5β (305–450): forward 5′-GGAATTCCATGTCTCTAAGCTCAGGGGC-3′, reverse 5′- GGAAT TCTTAGAGGATGGTCTCCAGCC-3′; FLAG-LL5β(381–510): forward 5′-GGAATTCCAT GTGTGGATCAATGGAGCTT-3′, reverse 5′-GGAATTCTTAAGAGCCCTTCCGG TG-3′; FLAG-LL5β(448–558): forward 5′-GGAATTCCATGACCATCCTCAGTCTCTGTGC-3′ reverse 5′-GGAATTCTTAGGTTGCTTTCAGAAACGC-3′; FLAG-LL5β(381–558): forward 5′-GGAATTCCATGTGTGGATCAATGGAGCTT-3′; reverse 5′-GGAATTCTTAGGTTGCT TTC AGA AACGC-3′; FLAG-LL5β(381–470): forward 5′-GGAATTCCATGTGTGGATCAA TGGAG CTT-3′; reverse 5′-GGAATTCTTACACCGTGGTGCCAG-3; FLAG-LL5β(425–558): forward 5′-GGAATTCCATGCACCGAAGACAGAGGGAG-3′; reverse 5′-GGAATTCTTAGG TTGCTTTC AG AAACGC-3′. Plasmids for His-ERC1-N and His-ERC1-C were prepared as described [13]. The cDNAs for fusion proteins GST-LL5β(381–510) and His-ERC1(270–370) were cloned into pGEX-4T-3 and pET-28b(+) plasmids, respectively (Genescript-DBA Italia).

### Antibodies and reagents

Rabbit polyclonal antibodies (pAbs) for ERC1, FLAG, FN, LL5β (recognizing human and monkey LL5β, but not mouse LL5β), and mouse monoclonal antibodies (mAbs) for ERC1a (clone ELKS-30), tubulin, FLAG clone M2 (Sigma-Aldrich); rabbit pAb for GFP (Life Technologies); chicken pAb for GFP and rabbit pAb (Abcam); rabbit pAb anti DS-RED (Clontech); rabbit pAb for Liprin-α1 (Protein Tech); mouse mAb for filamin A (Millipore), paxillin (clone 349, BD Biosciences), GST (Amersham Biosciences), His-tag (Qiagen), LL5α/β (clone 1H12) [19]; hamster mAb for LL5α [5] was kindly provided by dr. Yuko Mimori-Kiyosue (RIKEN

Center for Biosystems Dynamics Research, Kobe, Japan). Secondary Abs Alexa-488, Alexa-568, Alexa-546 and Alexa-647, phalloidin Alexa-568 and Alexa-647, and Oregon green 488-gelatin (Life Technologies); HRP-conjugated anti-rabbit and anti-mouse secondary Abs (Jackson and Amersham Biosciences). Other reagents included: FN (Corning); poly-L-lysine hydrobromide.

## Cell culture and transfection

COS7 green monkey kidney cells were cultured in DMEM with 10% fetal clone III (Hyclone). MDA-MB-231 human breast adenocarcinoma cells were grown in DMEM/F12 1:1 with 10% fetal bovine serum. NIH-3T3 mouse embryonal fibroblasts were cultured in DMEM with 10% fetal bovine serum. Cells on plastic or on round 24 mm diameter glass coverslips were transfected with Lipofectamine-2000 (Life Technologies). For 6-well plates 1–4 μg plasmid DNA in Opti-MEM transfection medium were used for each transfection. After 3.5–4 h the transfection medium was replaced with complete medium, and cells were processed 24–48 h later. All cells were regularly checked for mycoplasma contamination.

## Immunoprecipitation and immunoblotting

Cells were washed twice with ice-cold TBS (150 mM NaCl, 20 mM Tris-HCl pH 7.5), and lysed with 50–150 μl of lysis buffer (0.5% Triton X-100, 150 mM NaCl, 20 mM Tris-Cl pH 7.5, 1 mM NaV, 10 mM NaF, anti-proteases Complete 1× (Roche), 0.5 mM PMSF (Sigma-Aldrich) and 1 mM DTT). After 15 min at 4°C the insoluble material was removed by centrifugation at 16000 RCF for 10 min at 4°C. Protein concentration was determined using Bradford protein assay (Bio-Rad). Denatured lysates were separated by SDS-PAGE and transferred to 0.45 μm nitrocellulose membranes (GVS). Membranes were incubated with primary antibodies, HRP-conjugated secondary antibodies, and revealed by Clarity with ChemiDoc MP Imaging System (Bio-Rad). The uncropped blot images of the panels presented in the Figures are shown in S1 Raw images. Quantification of protein levels was done with ImageLab software (Bio-Rad). For reprobing, membranes were stripped by 5–10 min incubation at RT with 0.2 M glycine, 0.1% SDS, 1% Tween-20, pH 2.2, then washed at neutral pH before reprobing with the indicated antibodies. For immunoprecipitation cell lysates were incubated with Protein-A–Sepharose beads (Cytiva), Pierce Protein G Agarose (Thermo Scientific) conjugated to antibodies, GFP-Trap (Chromotek), or anti-FLAG-M2 Affinity Gel (Sigma-Aldrich) before processing for SDS-PAGE.

## Pulldown assays

Constructs were transformed into *Escherichia coli* BL21 (DE3) strain, and protein expression was induced by 0.25 mM IPTG for 2h at 37°C. Following resuspension in sonication buffer (20 mM Tris pH 8; 200 mM NaCl; 20% glycerol; 1 mg/ml Lysozyme (Sigma-Aldrich); 2 μg/mL DNase I (Sigma-Aldrich); 20 μg/mL RNase (Sigma-Aldrich); 10 mM imidazole; Complete 1x EDTA-free from Roche; 1 mM DTT), samples were lysed by sonication with a 3 mm titanium probe (TS 103, Bandelin Electronic). Lysates were clarified by ultracentrifugation at 35000 g for 30 min at 4°C. Glutathione-Agarose beads (Sigma-Aldrich) and Ni-NTA Agarose beads (Qiagen) were used for pull down experiments. Aliquots of beads (20 μl/sample) were incubated for 1 h on ice with supernatants from bacterial lysates (100 μl/sample). After incubation, beads were washed 3 times in washing buffer (20mM Tris-HCl pH 8, 150 mM NaCl, 10 mM imidazole, 0.1% Triton X-100, 1 mM DTT, Complete 1x). After washing, equal amounts of supernatant from bacterial lysates with the other fusion protein were added to the beads and

kept on ice for 30 min with resuspension every 5 min. Beads were washed and analyzed with unbound fractions by SDS-PAGE and immunoblotting.

## Saturation binding and competition assay

Overnight cultures of *Escherichia coli BL21* (DE3) cells transformed with His-ERC1(270–370) expressing plasmid were diluted 1:100 in LB media. At $A_{600}$ of 0.7, insert expression was induced with IPTG (0.5 mM) and grown overnight at 18˚C. Cells were sedimented and lysed by sonication (6 sec on/off for 7 cycles) in 20 mM Tris-HCl pH 7.5, 300 mM NaCl, 5 mM imidazole, and protease inhibitor cocktail. After sonication lysate was centrifuged at 16000 g for 30 min at 4˚C. Supernatant was incubated with Ni-NTA Agarose Beads (Qiagen) for 2.5 h at 4˚C. Beads were washed with buffers 1 and 2 (20 mM Tris pH 7.3, 300 mM NaCl) containing increasing concentrations of imidazole (buffer 1: 20 mM; buffer 2: 50 mM). Protein was eluted in 20 mM Tris-HCl pH 7.5, 300 mM NaCl plus 300 mM imidazole, dialyzed in 20 mM HEPES pH 7.5, 150 mM NaCl and stored at -80˚C. Overnight cultures of *E.coli* BL21 transformed with GST-LL5β(381–510) were diluted 1:100 in LB media. At $A_{600}$ of 0.7, insert expression was induced with IPTG (0.5 mM) and grown overnight at 30˚C. Cells were sedimented and lysed by sonication (6 sec on/off for 7 cycles) in 50 mM Tris-HCl pH 8.5, 150 mM NaCl with protease inhibitor cocktail. After sonication, the lysate was centrifuged at 16000g for 30 min at 4˚C. Supernatant was incubated with Glutathione Agarose Beads (GST, from Pierce) for 2.5 h at 4˚C. Beads were washed twice with 50 mM Tris-HCl pH 8.6, 150 mM NaCl. Protein was eluted in 50 mM Tris pH 8.6, 150 mM NaCl and 10 mM of reduced glutathione, dialyzed in 20 mM HEPES pH 7.5, 150 mM NaCl, 0.2 mM Tris(2-carboxyethyl)phosphine reducing agent (considering cysteine presence in construct sequence) and stored at -80˚C. For the isolation of tag-free GST-LL5b(381–510): 1 mg of GST-LL5β(381–510) recombinant protein was incubated overnight under shaking conditions at room temperature with 100 μl of thrombin–agarose beads (Merck, RECOMT-1KT) in cleavage buffer. The supernatant was collected and incubated twice with GSH resin to remove the GST tag. After centrifugation of GSH resin, tag-free GST-LL5β(381–510) was obtained.

Amplified Luminescent Proximity Homogenous Assay (ALPHA) was used to study the interaction between ERC1(270–370) and LL5β(381–510) [23]. The assay was performed in 384-well white OptiPlates (PerkinElmer) in a final volume of 25 μl and optimized by titrating both interacting partners (to determine the optimal protein: protein ratio). Values out of the "hooking zone", where quenching of the signal is due to an excess of the binding partner, were determined for the optimal concentrations of probe and protein. All reagents were tested in the nanomolar range in buffer A (25 mM HEPES pH 7.4, 100 mM NaCl, 0.1% BSA). For the optimization of the assay, three concentrations of His-ERC1(270–370) (5, 15, 45 nM) were incubated with increasing concentrations of GST-LL5β(381–510) (0–400 nM) for 30 min. Subsequently, anti-GST-Acceptor beads (PerkinElmer, code AL109C) (20 μg/ml final concentration) and Anti-6xHis Alpha Donor Beads (PerkinElmer code AS116D, 20 μg/ml final concentration) were added, and the reaction was incubated at room temperature in the dark for 45 min, 1 h, 2 h, 2.5 h and 3 h to reach equilibrium. Signal intensity was measured using EnSight multimode plate reader (PerkinElmer, HH34000000) and signal intensity was quantified by subtracting the signal of the background, calculated in the absence of the protein and/ or of the probe and with protein elution buffer only (nonspecific binding). Assays were performed in quadruplicate with different protein preparations. Apparent equilibrium dissociation constants (app $K_d$) were determined using nonlinear regression fits of the data according to a one-site binding model in GraphPad Prism®, version 5.0 (GraphPad Software, Inc.). Fitting values have been reported as averaged mean ± standard deviation of all the experiments.

For the competitive assay, His-ERC1(270–370) (45 nM), was incubated with GST-LL5β(381–510) (100 nM) and the Tag-free LL5β(381–510) protein at different concentrations (200 nM, 100 nM, 50 nM, 25 nM, 12.5 nM, 3.125 nM, 1.57 nM) for 45 min at room temperature. Acceptor and donor beads were then added, incubated for 1 h, and fluorescence intensity measured. $IC_{50}$ was calculated using nonlinear regression fits of the data in GraphPad Prism$^{®}$.

## NMR experiments

Preparation of samples for NMR spectroscopy was as follows. Overnight cultures of bacteria transformed with either plasmid were diluted in 1 liter of M9 minimal medium (48.5 mM $Na_2HPO_4$, 22.0 mM $K_2HPO_4$, 8.5 mM NaCl, 0.2 mM $CaCl_2$, 2.0 mM $MgSO_4$, 1 mg/liter each of biotin and thiamine, 7.5 mM $(NH_4)_2SO_4$ or $(^{15}NH_4)_2SO_4$, and 11.1 mM D-glucose) containing ampicillin. Cultures were grown at 37˚C with shaking until OD600 = 0.8 was reached. Protein expression was induced by 0.25 mM IPTG for 2 hours at 37˚C. Cell pellets were resuspended in 100 ml of sonication buffer (20 mM Tris pH 8, 200 mM NaCl, 20% glycerol, 1 mM DTT, 2 μg/ml DNase I (Sigma), 20 μg/ml RNase (Sigma), Complete 1x EDTA-free (Roche). Cells were lysed by sonication and lysates were centrifuged at 35000 g for 30 min at 4˚C. Supernatant was loaded on 2 x 5 ml GSTrapTM FF columns (Cytiva) equilibrated with binding buffer (PBS, 50 mM DTT, pH = 7.3). After washing with binding buffer, the column was equilibrated with thrombin cleavage buffer and 10 ml of 1 U/μl thrombin (Cytiva 27-0846-01) were added and incubated at room temperature overnight. The next day cleaved LL5β(381–510) was recovered by washing the column with PBS. The uncleaved protein and the cleaved GST-tag were eluted with 50 mM Tris-HCl, 10 mM reduced glutathione, pH 8. The tag-free protein solution was concentrated to 2.5 ml and loaded on PD-10 desalting column. LL5β(381–510) was eluted with PBS at pH 7.2 or 20 mM sodium phosphate, 150 mM NaCl, pH 7.2 or 20 mM Tris, 150 mM NaCl, pH 7.2, and concentrated. 10% $D_2O$ was added for lock signal.

Protein expression protocol for His-ERC1(270–370) was similar to the one for GST-LL5β (381–510). Cell pellets were resuspended in 100 ml of sonication buffer with 10 mM imidazole. The supernatant obtained after sonication was loaded on a 5 ml HisTrapTM FF column (Cytiva) and the protein was eluted by a linear gradient of 0–100% elution buffer (sonication buffer + 500 mM imidazole). The protein solution was concentrated to 2.5 ml and loaded on PD-10 desalting column. His-ERC1(270–370) was eluted with 20 mM sodium phosphate, 150 mM NaCl, pH 7.2 or 20 mM Tris, 150 mM NaCl, pH 7.2 and concentrated. 10% $D_2O$ was added for lock signal. These procedures were followed for both unlabeled and $^{15}N$-labeled samples.

1D $^1H$ experiments on LL5β(381–510) and His-ERC1(270–370) were recorded on Bruker AVANCE III operating at 950.20 or Bruker NEO operating at 899.92 MHz equipped with a cryogenically cooled triple-resonance probehead (TCI) and using standard pulse sequences. The latter spectrometer was used to record 2D HN correlation experiments of $^{15}N$-LL5β(381–510) alone and on the $^{15}N$-LL5β(381–510):His-ERC1(270–370) adduct (1:1 molar ratio). 1D $^1H$ experiments on LL5β(381–510) were recorded at various temperatures in the range 278–308 K. All the other spectra were recorded at 298 K.

Sensitivity Improved $^1H$-$^{15}N$ HSQC experiments [24] were acquired with an interscan delay of 1.2 s, with sweep widths of 13 ppm ($^1H$) × 35 ppm ($^{15}N$) and 2048 × 512 points in the two dimensions. $^{15}N$ pulses were given at 118.0 ppm and the $^1H$ carrier was placed at 4.7 ppm. Decoupling of $^{15}N$ was achieved with garp (250 μs) decoupling sequences [25]. The $^1H$-$^{15}N$ Fast-HSQC experiments [26] were recorded with a wider spectral window to observe the signals of the side chain of arginine residues. These spectra were acquired with 16 scans, with an

interscan delay of 1 s, with sweep width of 12 ppm ($^1$H) × 80 ppm ($^{15}$N) and 2048 x 512 points in the two dimensions. $^1$H and $^{15}$N carrier frequencies were 4.7 ppm and 95 ppm respectively. All gradients employed had a smoothed square shape.

All the spectra were acquired, processed, and analyzed by using the spectrometer's Bruker TopSpin software.

### Immunofluorescence and image analysis

Cells grown on FN-coated glass coverslips were washed and fixed for 10 min at RT in 3% para-formaldehyde in PBS+ (phosphate buffer saline with $Ca_2^+$ and $Mg_2^+$). After washing and quenching for 10 min with 50 mM $NH_4Cl$ in PBS+, cells were permeabilized for 10 min with 0.1% Triton X-100. Coverslips were incubated for 2h with primary antibodies, for 45 min with secondary antibodies (RT), then mounted in ProLong Gold Antifade Reagent (Life Technologies). Wide field images were acquired with Zeiss Axio Observer Z.1 with Hamamatsu EMCCD 9100–02 camera equipped with Plan-Apochromat 63× (NA 1.4) lens. Confocal images were acquired with Perkin Elmer UltraVIEW spinning disk confocal microscope equipped with EM-CCD camera, with Plan-Apochromat 63× (NA 1.4) lens; with Leica TCS SP8 SMD FLIM laser scanning confocal microscope equipped with HC PLAPO CS2 63× (NA 1.4) lens; or with Leica TCS SP5 laser scanning confocal microscope equipped with HCX PLAPO λ blue 63X (NA 1.4) lens. For the quantification of PMAPs, protrusions were identified as cellular extensions with clear F-actin-positive edge. The percentage of PMAP–positive protrusions was evaluated in 10–21 cells for each experiment. Colocalization of proteins with GFP-ERC1 in condensates was evaluated by the Pearson's correlation coefficient on confocal images [27]. The coefficient (1 = perfect correlation, 0 = no correlation, -1 = perfect anti-correlation) was calculated using the plugin Colocalization Finder of ImageJ. For quantification, 3–4 GFP-ERC1–positive condensates were randomly picked from 6–10 cells per condition per experiment.

### Cell motility assays

For cell spreading, MDA-MB-231 cells were re-plated 24 h after transfection on coverslips coated with 10 μg/ml FN. After 18 h cells were processed for immunofluorescence. Projected cell area was measured with ImageJ (n = 30–50 cells per condition per experiment). For 2D random migration assays, MDA-MB-231 cells were replated on FN coated 6-well plates (2.5 μg/ml, overnight at 4°C) 24 hours after transfection. After culture overnight, cells were washed with PBS and supplied with fresh medium. The cells were imaged for 5 h (one frame every 10 min) with IncuCyte Live-Cell Imaging System equipped with 10x lens (Sartorius). For 3D random migration assay, 3D matrices were prepared as published [28] from NIH-3T3 fibroblasts. Quality of matrices were detected by immunofluorescence with anti-FN Ab. Z-stacks acquisition was used to evaluate their thickness. For the assays, MDA-MB-231 cells were re-plated on 3D matrix-coated 6-well plates 24 h after transfection, and cultured over-night. Cells were then washed and supplied with fresh growth medium. The cells were imaged every 10 min for 8 h with IncuCyte Live-Cell Imaging System equipped with 10x lens. The pathway of transfected cells in 2D and 3D migration assays was tracked and analyzed with Image J (plugins Manual tracking and Chemotaxis tool). About 70–100 cells per condition were analyzed in each experiment.

### Matrigel invasion assays

Clones from MDA-MB-231 cells expressing either GFP, GFP-LL5β(381–510), or GFP-LL5β (305–450) were obtained as described. Growth curves were obtained by plating MDA-MB-231

cells stably transfected with the indicated GFP-tagged constructs in 96-well plates (2500 cells/well). Cells were incubated at 37˚C and cultured for up to 5 days. MTT assays were performed daily in order to assess cells proliferation. MDA-MB-231 (100000 cells in 100 μl) were seeded on Matrigel-coated 8 μm-pore polycarbonate membrane transwells (Corning) in DMEM/F12 medium containing 0.1% BSA (Sigma-Aldrich). NIH-3T3-conditioned medium in the lower chamber was used as chemoattractant. After 5 h of incubation at 37˚C non-invading cells were removed from the upper side with a cotton swab, and cells invading and crossing the membranes were fixed with PFA 3% and stained with DAPI.

## Extracellular matrix degradation assay

Gelatin degradation assay was performed as published [17, 29]. Glass coverslips coated for 1 h at RT with 0.5 mg/ml poly-L-lysine (Sigma-Aldrich) were quenched 15 min at 4˚C with 0.5% glutaraldehyde in PBS, and then coated for 10 min at RT with Oregon–green–conjugated gelatin (Life Technologies) diluted 1:4 in 0.2% gelatin in PBS. Finally, the coverslips were coated with 10 μg/ml FN in PBS for 1 h at 37˚C. The cells were transfected for 24 h and re-plated on the gelatin-coated coverslips. The cells were left to spread for 5 h and processed for immunofluorescence. The dark areas of gelatin degradation and the projected cell areas were quantified by ImageJ on thresholded images. Invadopodia were identified by immunostaining for phalloidin. 30–50 cells per condition per experiment were analyzed.

## Statistical analysis

Statistical analysis was performed using GraphPad Prism 9.0. All datasets were tested for normality using Shapiro-Wilk test. For datasets with normal distribution, the statistical significance was calculated using unpaired two-tailed Student's t test or one-way ANOVA with Dunnett's or Tukey's posthoc. For datasets with non-normal distribution, the statistical significance was calculated using Kruskal-Wallis test with Dunn's posthoc. Data are presented as mean ± SEM.

## Results and discussion

### Identification of the minimal regions required for the interaction of ERC1 with LL5β

We have set to identify the interacting protein fragments to be used to interfere with PMAPs formation and/or function, with the aim of showing that the interaction between ERC1 and LL5β is important for tumor cell motility and invasion. Regions of interaction between LL5β (residues 306–562) and ERC1 (residues 200–400) were identified by *in vitro* pulldown assays [12]. We attempted to identify smaller protein regions involved in the intermolecular interaction in cells. The mCherry-LL5β full length protein interacts with GFP-tagged ERC1 and ERC1-N, weakly with ERC1-Δ147, while did not interact with ERC1-C (Fig 1A). In MDA-MB-231 cells both LL5α and LL5β are expressed, contributing to the motility and invasion of these cells [7]. In COS7 cells the endogenous LL5 protein is represented mostly by LL5α (S1A Fig), which intriguingly was not co-immunoprecipitated with any of the ERC1 constructs tested (Fig 1A), possibly due to a relatively weak interaction between the two proteins in the cell. To define the minimal polypeptide regions required for the interaction of LL5β and ERC1 we have produced and tested by co-immunoprecipitation several fragments of LL5β. We started with an ERC1 fragment (ERC1(199–402)) previously identified as interacting with LL5β [12]. We found that ERC1(199–402) interacts efficiently with different LL5β fragments (Fig 1B, including a summary from different co-immunoprecipitation experiments),

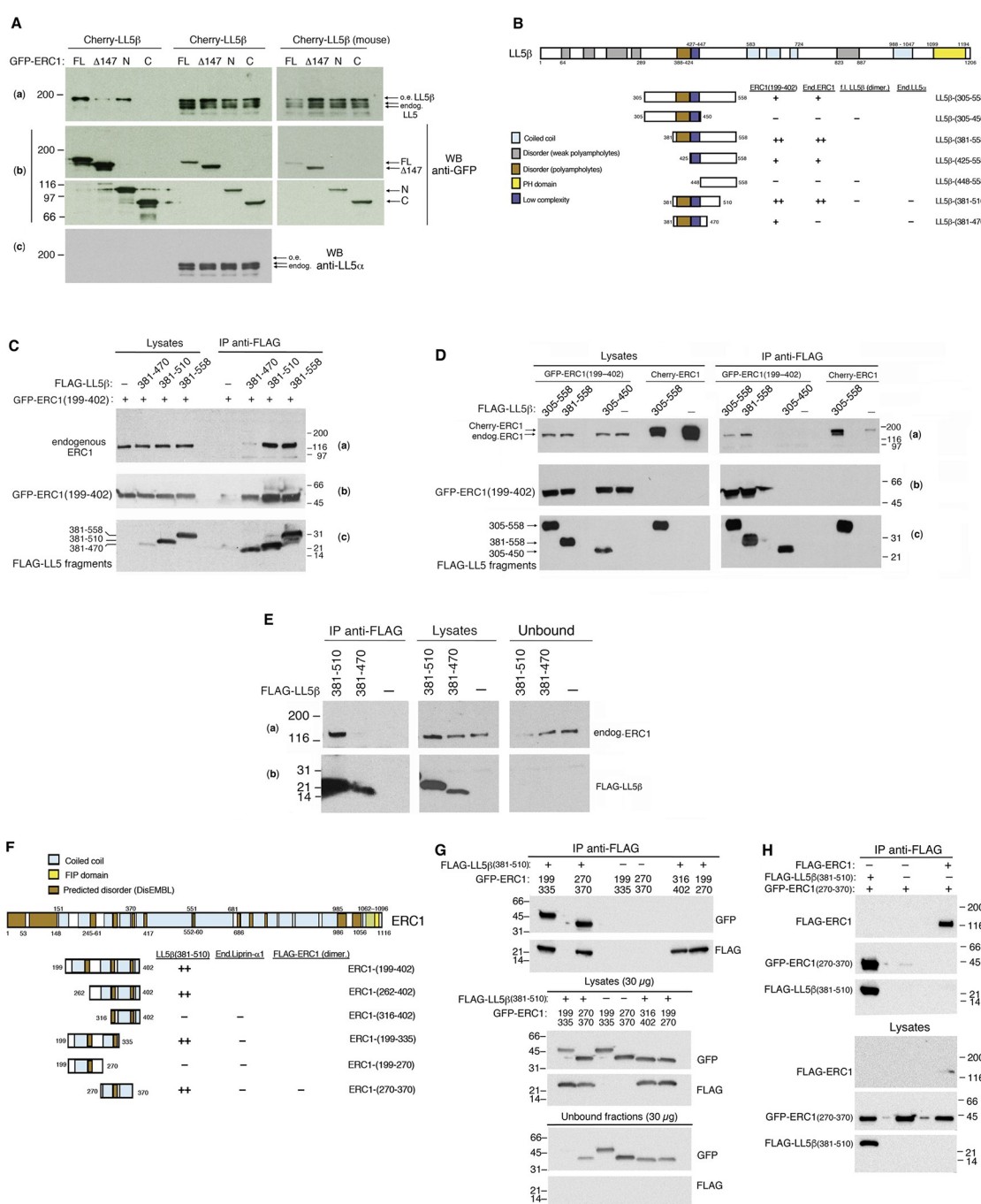

**Fig 1. Identification of the minimal fragments required for the interaction of ERC1 with LL5β.** (**A**) Aliquots (300 μg protein) of lysates from COS7 cells transfected with the indicated GFP–tagged ERC1 constructs were immunoprecipitated with anti-GFP Abs. Immunoprecipitates (IP), lysates (Lys, 30 μg) and unbound fractions (Ub, 30 μg) were immunoblotted to detect the GFP–tagged ERC1 constructs (**b**), or incubated with anti-LL5 1H12 mAb detecting both mCherry-LL5β and the endogenous LL5 polypeptides (**a**). The upper filters were reblotted with an antibody specific for LL5α (filter **c**). (**B**) Scheme of the constructs and summary of the results of co-immunoprecipitation experiments to identify the fragment LL5β(381–510) as an efficient ERC1-binding region. (**C-D**) Aliquots (300 μg protein) of lysates of COS7 cells co-transfected with either GFP-ERC1(199–402) or mCherry-ERC1 together with FLAG-tagged LL5β constructs, were immunoprecipitated with anti-FLAG mAb. Immunoprecipitates (IP) and lysates (Lys, 30 μg) were immunoblotted to detect endogenous ERC1 (**a**), GFP–ERC1(199–402) (**b**), and FLAG-LL5β fragments (**c**). (**E**) Aliquots (300 μg protein) of lysates of COS7 cells transfected with FLAG-tagged LL5β constructs were immunoprecipitated with anti-FLAG mAb. Immunoprecipitates (IP), lysates (Lys, 30 μg) and unbound fractions (30 μg) were immunoblotted to detect endogenous ERC1 (**a**) and FLAG-LL5β fragments (**b**). (**F**) Scheme of the constructs and summary of the results of co-immunoprecipitation experiments to identify the fragment ERC1(270–370) as an efficient LL5β-binding region. Different colors indicate predicted coiled

coils (by the COIL program), a domain of family of Rab11 interacting protein (FIP), and predicted intrinsically disordered regions (by the DisEMBL program, http://dis.embl.de) [32]. (**G**) Aliquots (150 μg protein) of lysates of COS7 cells co-transfected with FLAG-LL5β(381–510) and GFP-tagged ERC1 constructs were immunoprecipitated with anti-FLAG mAb. Immunoprecipitates (IP), lysates (30 μg) and unbound fractions (30 μg) were immunoblotted with anti-GFP and anti-FLAG Abs. (**H**) GFP-ERC1(270–370) does not interact with FLAG-ERC1. Immunoprecipitation with anti-FLAG Abs from lysates (150 μg of protein/IP) of COS7 cells transfected with the indicated constructs. Lysates, 30 μg. Numbers to the left of blots in **A**, **E** and **G**, and to the right of blots in **C**, **D** and **H** indicate molecular weight standatds (kD).

including LL5β(381–510) that was identified as the shortest region of LL5β interacting efficiently also with the endogenous ERC1 protein (Fig 1C and 1D). Of note, immunoprecipitation of endogenous ERC1 with an antibody raised against amino acid residues 20–40 of ERC1 could not reveal the interaction with LL5β(381–510) (not shown), possibly due to interference with the formation of the ERC1-LL5β complex due to binding of this Ab to the amino-terminal region of ERC1. On the other hand, immunoprecipitation from cells transfected with Flag-LL5β fragments showed that endogenous ERC1 was specifically and efficiently coprecipitated by LL5β(381–510) (Fig 1E). The LL5β(381–510) region includes a predicted intrinsically disordered region (IDR) at residues 388–424, identified by the MobiDB program for prediction of intrinsic protein disorder [30], and a low complexity region (LCR) including residues 427–447, identified by the program SEG [31] (Fig 1B). Of note, the overlapping fragment LL5β (305–450) including the same IDR and LCR present in LL5β(381–510) did not interact with ERC1(199–402) nor with endogenous ERC1 (Fig 1D), and was used as a negative control in the next experiments.

No data are available on the dimerization of LL5β. The formation of LL5β dimers is suggested by the presence of predicted coiled coil regions (Fig 1B), including a short predicted coiled coil region coinciding with the LCR (residues 427–447) in LL5β(381–510). Co-immunoprecipitation experiments showed that the full length LL5β protein was able to interact with full length LL5β with a different tag. In contrast, the short LL5β(381–510) fragment as well as other fragments without predicted coiled coils could not coprecipitate with full length LL5β (S1B Fig), suggesting that these fragments do not dimerize with endogenous LL5β. Therefore, we conclude that the interaction of LL5β(381–510) with ERC1 does not require LL5β dimerization.

A second set of coimmunoprecipitation experiments was performed to restrict the region of interaction of ERC1 binding to LL5β. A number of GFP–tagged ERC1 constructs were co-transfected with FLAG-LL5(381–510) in COS7 cells (Fig 1F, including a summary from different co-immunoprecipitation experiments). We restricted to ERC1(270–370) the region of ERC1 required for efficient binding to LL5β (Fig 1G).

The amino-terminal region of ERC1 is involved in homodimerization [33]. We tested whether ERC1(270–370) was sufficient for homodimerization. While ERC1(270–370) interacted efficiently with LL5β(381–510), we could not detect an interaction of this fragment with the full length ERC1, suggesting that ERC1(270–370) is monomeric (Fig 1H) and that a dimeric form of ERC1 is not required for binding to LL5β. In addition, ERC1(270–370) is part of the larger fragment of ERC1 (residues 126–567) required for the interaction with Liprin-α1 [34], but was unable to coimmunoprecipitate with endogenous Liprin-α1 (S1C Fig).

To further characterize the interaction between the two fragments, we cloned and expressed in *E. coli BL21* the interacting regions of the ERC1 and LL5β proteins. The specificity of the interaction between the two fragments was confirmed by pulldown of GST-LL5β(381–510) on beads prebound to His-ERC1(270–370) (Fig 2A), and by the complementary pulldown of His-ERC1(270–370) on beads prebound to GST-LL5β(381–510) (Fig 2B). In addition, we used the bead-based ALPHA Screen to quantitatively measure the interaction. His-ERC1(270–370) and

GST-LL5β(381–510) protein fragments were expressed and purified. Protein purification was evaluated by Coomassie staining and fractions with similar purity were merged (Fig 2C). In saturation binding experiments 5, 15, and 45 nM of His-ERC1(270–370) was titrated with an increasing amount of GST-LL5β(381–510); after incubation donor and acceptor beads were added to the reaction mixture and the fluorescence signal was measured. In these conditions, we calculated three different apparent dissociation constants ($K_d$) in the nanomolar range (Fig 2D): $Kd_{(45nM)}$ = 99.78 nM ($R^2$ = 0.92); $Kd_{(15\ nM)}$ = 109.5 nM ($R^2$ = 0.90); $Kd_{(5\ nM)}$ = 78.75 nM ($R^2$ = 0.92).

To investigate the reversibility of the reaction we performed a competition assay using the untagged LL5β(381–510) fragment. We incubated the tagged interactors at the concentration in which we obtained the highest signal before the hook point drop, in the presence of increasing concentration of the untagged LL5β(381–510) fragment. After the addition of the beads, we observed a progressive decrease of the signal intensity indicating the dynamic nature of the interaction (Fig 2E).

NMR has become the preferred technique for obtaining atomic resolution information also in the presence of highly flexible proteins. The $^1$H NMR spectra of His-ERC1(270–370) and LL5β(381–510) (Fig 3A and 3B) show the typical features of a disordered proteins, with very limited signals dispersion and clustering of the signals around the typical chemical shift of the various amino acids. We used $^1$H-$^{15}$N HSQC spectra to assess the interaction between the two constructs. Comparing the $^1$H-$^{15}$N HSQC spectrum of LL5B(381–510) with the one of the 1:1 adduct with ERC1(270–370) (Fig 3C and 3D) it can be noted that some signals become broad beyond detection while others are shifted. These preliminary results confirm the presence of an interaction between the two constructs with a dissociation constant in the nanomolar range. This interaction is driven by a subset of residues of the two proteins that are yet to be identified. For this purpose, further analysis is needed when the residue-specific assignment will be available.

## LL5β(381–510) and ERC1(270–370) prevent the localization of endogenous ERC1 at PMAPs in migrating MDA-MB-231 cells

Expression of the ERC1-binding construct Flag-LL5β(381–510) in MDA-MB-231 breast cancer cells showed a diffuse cytoplasmic localization of the protein fragment, and induced the displacement of endogenous ERC1 which could not be detected at PMAPs at the protruding edge of MDA-MB-231 cells migrating on FN (Fig 4A). Quantification of the diffuse localization of the endogenous ERC1 protein in most cells expressing Flag-LL5β(381–510) suggests that the ERC1–binding fragment of LL5β can displace endogenous ERC1 from the protruding cell edge of migrating cells (Fig 4B). On the other hand, the control fragment Flag-LL5β(305–450) that does not interact with ERC1, did not affect ERC1 localization at PMAPs, nor did it colocalize with ERC1 at the cell edge (Fig 4A and 4B). The observed localization of FLAG-LL5β(305–450) at the edge of protrusions (lamellipodia) (Fig 4A) may be driven by binding of this fragment to the actin-binding protein filamin, since FLAG-LL5β(305–450) includes the previously identified filamin binding region of LL5β, corresponding to residues 323–356 of human isoform a of LL5β, highly conserved in mouse LL5β [7]. Overexpression of full length LL5β appeared to increase the localization of endogenous ERC1 at PMAPs in migrating tumor cells (not shown), in support of the previous observation that LL5 proteins are required for the formation of ERC1–positive PMAPs: indeed, silencing of endogenous LL5 proteins in migrating MDA-MB-231 cells is known to cause delocalization of ERC1 from protrusions [7]. While endogenous ERC1 was affected by the expression of ERC1-binding FLAG-LL5β(381–510), this fragment did not displace endogenous Liprin-α1 or LL5 proteins

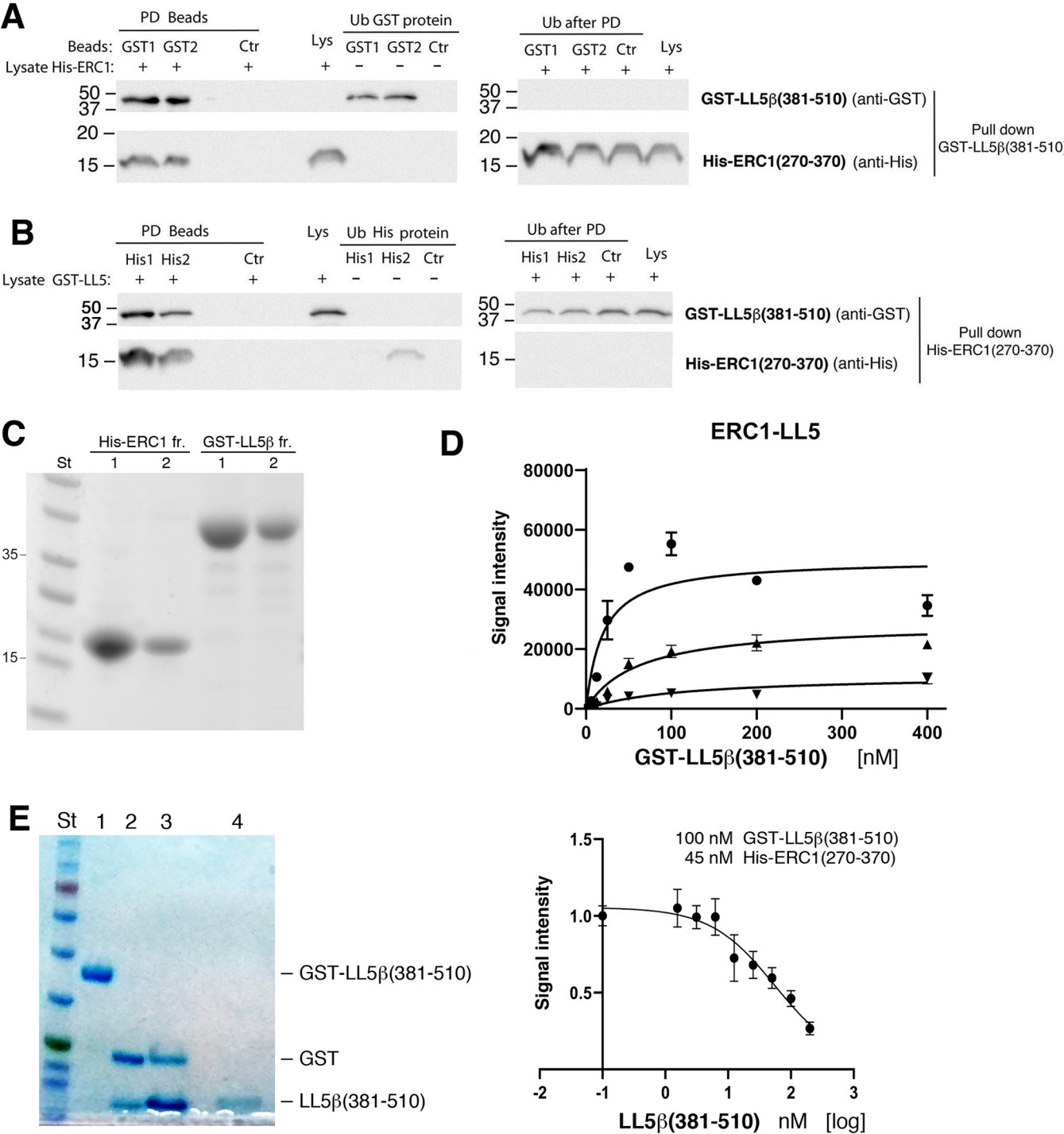

**Fig 2. Direct interaction between ERC1 and LL5β *in vitro*.** (**A**) Glutathione empty (Ctr) or GST-LL5β(381–510)–bound agarose beads were incubated with equal amounts of His-ERC1(270–370). Pulldown (PD) and unbound fractions (Ub) were immunoblotted to detect GST-LL5β(381–510) (upper filters) or His-ERC1(270–370) (lower filters). (**B**) Ni-NTA-Agarose empty (Ctr) or His-ERC1(270–370)–conjugated beads were incubated with bacteria lysates with equal amounts of GST-LL5β(381–510). PD and Ub fractions were blotted for the detection of the indicated antigens. (**C**) Coomassie staining showing purification results and purity of recombinant His-ERC1 (270–370) and GST-LL5β(381–510). His-ERC1: lane 1, pool of eluted fractions 1–3 of His-ERC1(270–370) after affinity purification and dialysis; lane 2, pool of eluted fractions 4–5 after dialysis. GST-LL5β: lane 1, pool of eluted fractions 5–7 of GST-LL5β(381–510); lane 2, pool of eluted fractions 8–12 after dialysis. (**D**) ALPHA screen assays to test *in vitro* interaction of His-ERC1(270–370) and GST-LL5β(381–510). Saturation binding experiments with 5, 15 and 45 nM of His-ERC1 and a titration of GST-LL5β (6.25 nM, 12.5 nM, 25 nM, 50 nM, 100 nM, 200 nM and 400 nM). To calculate the equilibrium dissociation constant (Kd) we considered GST-LL5β values below 100 nM, Kd was determined from nonlinear regression fits of the data according to a one-site binding model. Means ±SD derived from duplicates. (**E**) Left: preparation of tag-free LL5β(381–510). Coomassie staining after

SDS-PAGE. Lanes are: 1, LL5β(381–510); 2, GST and LL5β(381–510) after digestion with thrombin; 3–4, LL5 fragment after the first and second incubation with Glutathione resin, respectively. Right: competition binding experiments with 45 nM of His-ERC1(270–370) and 100 nM of GST-LL5β(381–510) with titration of tag-free LL5β(381–510) (gel on the left) at the following concentrations: 0, 1.57nM, 3.125, 12.5 nM, 25 nM, 50 nM, 100 nM, nM 200 nM. IC$_{50}$ = 55.47 nM was determined from nonlinear regression fits of the data according to a one-site binding model. Mean ±SD values derive from three technical replicates. R$^2$ = 0.9.

from PMAPs in migrating MDA-MB-231 cells (S2A and S2B Fig). On the other side, the LL5β–binding fragment ERC1(270–370) had a diffuse cytoplasmic distribution, did not localize at PMAPs, and inhibited specifically the localization of endogenous ERC1 at PMAPs in protrusions of migrating MDA-MB-231 cells (Fig 4C and 4D). As observed for FLAG-LL5β (381–510), ERC1(270–370) had no evident effects on the localization at PMAPs of endogenous Liprin-α1 and LL5 proteins (S3A and S3B Fig). The results show that it is possible to interfere with the localization of endogenous components of the PMAPs at the front of migrating cells by expressing fragments that may interfere with the interaction between endogenous PMAP components ERC1 and LL5.

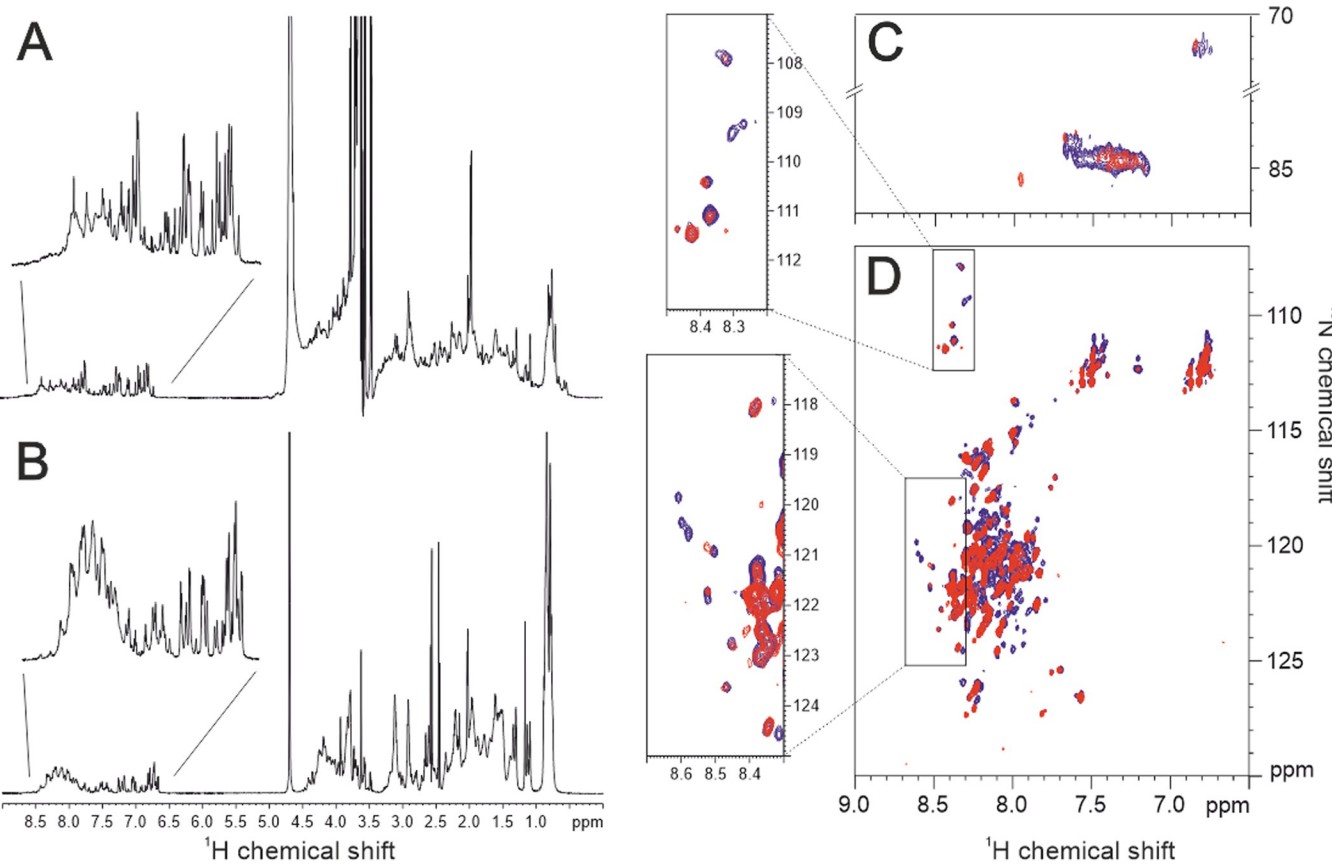

**Fig 3. NMR of ERC1(270–370) and LL5β(381–510).** (**A**) 900 MHz $^1$H NMR spectrum recorded on a 120 μM ERC1(270–370) sample in Tris-HCl 20 mM, NaCl 150 mM, pH 7.2 at 298K. (**B**) 950 MHz $^1$H NMR spectrum recorded on a 200 μM $^{15}$N LL5β(381–510) sample in phosphate buffer 10 mM, NaCl 150 mM, pH 7.2 at 298 K. (**C**) and (**D**) portions of Sensitivity Improved $^1$H-$^{15}$N HSQC spectra recorded on a 60 μM $^{15}$N LL5β(381–510) sample in Tris-HCl 20 mM, NaCl 150 mM, pH 7.2 alone (blue contours) and in the presence (red contours) of 60 μM ERC1(270–370) to monitor the changes occurring upon interaction. Spectrum (**C**) is a $^1$H-$^{15}$N Fast-HSQC recorded with a wide spectral width to detect the signals of arginine residues side chain. The upper panel on the left of spectrum (**D**) highlights the changes occurring in the region where the glycine residues cross peaks resonate. The lower panel shows the various effects of the two proteins' interaction on the spectrum, such as signal broadening and chemical shift perturbation.

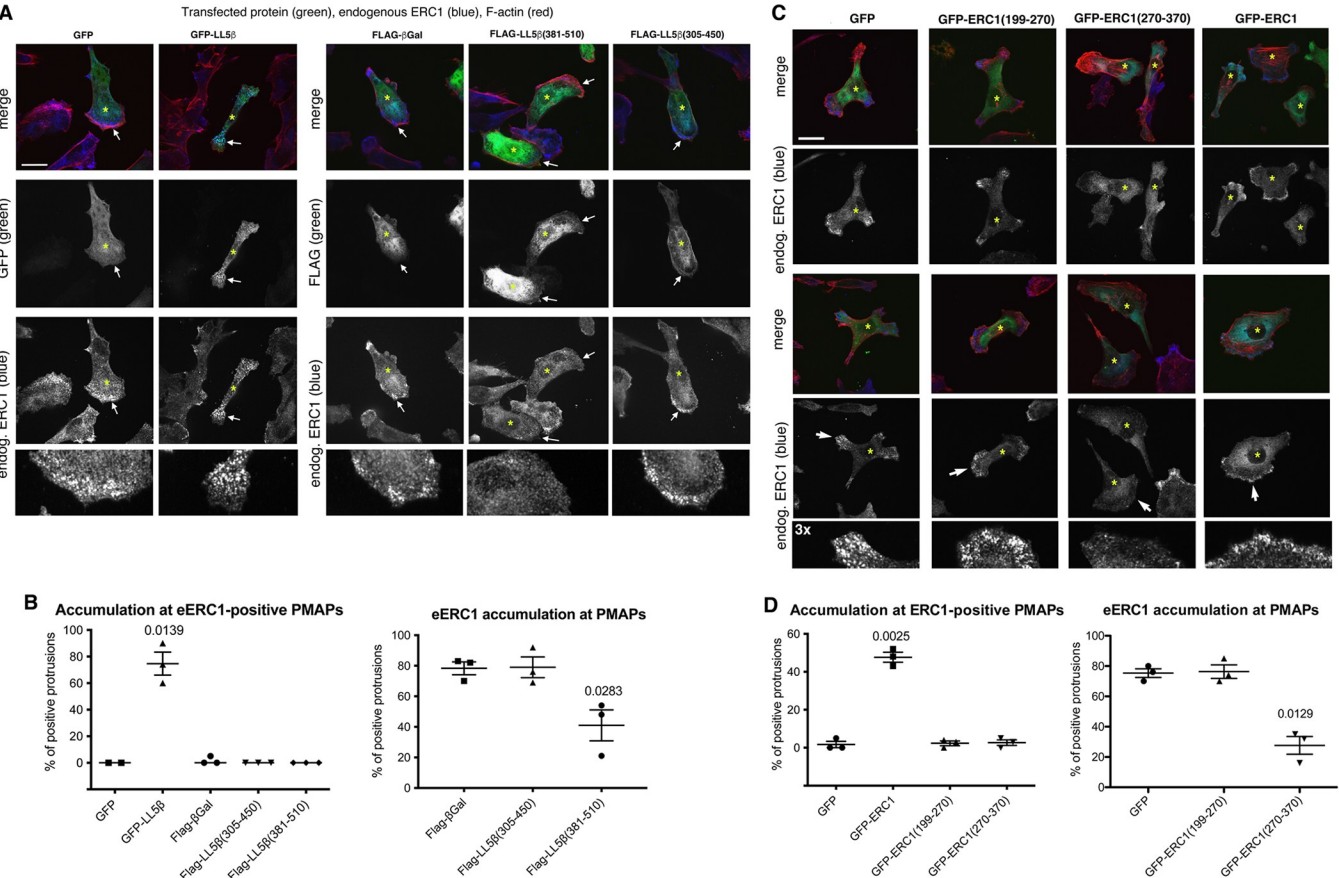

**Fig 4. Expression of either ERC1(270–370) or LL5β(381–510) displaces endogenous ERC1 from PMAPs at the edge of migrating MDA-MB-231 cells. (A)** Effects of the expression of GFP-tagged full length or fragments of ERC1 (green) on the localization of endogenous ERC1 (blue) in migrating MDA-MB-231 cells. Bar = 20 μm. Asterisks indicate transfected cells. Arrows indicate protrusions that are shown at 3x magnification in the bottom panels. **(B)** Accumulation of the transfected construct at PMAPs in protrusions (left graphs), and effects on the localization of endogenous ERC1 at PMAPs (right graphs) in cells expressing the indicated GFP-tagged proteins. mean ±SEM, n = 2–3 experiments. Left graph: one way ANOVA, Kruskal Wallis Test. Multiple comparisons. Uncorrected Dunn's test. Right graph: one way ANOVA, Tukey's multiple comparison test, with a single pooled variance. **(C)** Effects of the expression GFP-tagged full length or fragments of LL5β (green) on the localization of endogenous ERC1 (blue) in migrating MDA-MB-231 cells. Bar = 20 μm. **(D)** Accumulation of the transfected construct at PMAPs in protrusions (left graphs), and effects on the localization of endogenous ERC1 at PMAPs (right graphs) in cells expressing the indicated GFP-tagged proteins. Means ±SEM, n = 3 experiments; Brown-Forsythe and Welch ANOVA, Dunnet post-hoc vs GFP.

## LL5β(381–510) interferes with the binding of ERC1 to LL5β

We next asked whether the ERC1–binding fragment LL5β(381–510) was able to interfere with the binding between full length ERC1 and LL5β proteins. For this, we co-trasfected COS7 cells (which express endogenous ERC1 but very little if any endogenous LL5β) with GFP–tagged full length LL5β together with either the ERC1–interacting fragment FLAG-LL5β(381–510), the non-interacting fragment FLAG-LL5β(305–450), or the control non-interacting protein FLAG-βGalactosidase. Lysates from co-transfected cells were used to immunoprecipitate GFP-LL5β. Immunoblotting after immunoprecipitation revealed the association of endogenous ERC1 to full length GFP-LL5β. We observed an evident decrease of the co-precipitating endogenous ERC1 protein in immunoprecipitates from lysates of cells co-expressing the ERC1–interacting fragment FLAG-LL5β(381–510) when compared to cells co-expressing GFP-LL5β and βGalactosidase (Fig 5A and 5B). The association between endogenous ERC1 and full length GFP-LL5β was not affected in cells expressing the control fragment FLAG-LL5β(305–450), unable to interact with ERC1 (Fig 1D). The analysis of the

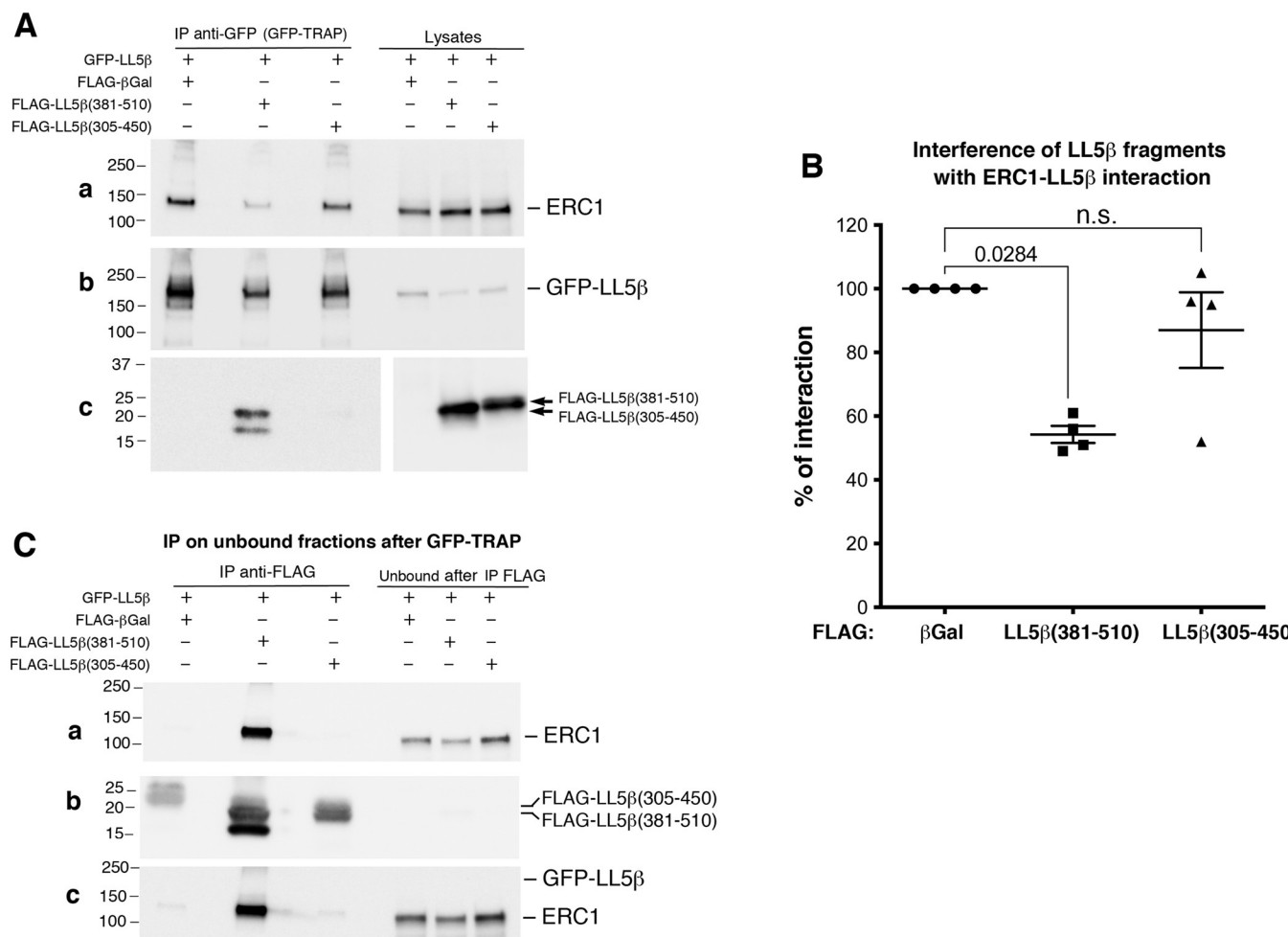

**Fig 5. The ERC1-binding fragment LL5β(381–510) interferes with the interaction between full length LL5β and ERC1 proteins.** COS7 cells were co-transfected with GFP-LL5β together with either FLAG-tagged βGalactosidase, LL5β(381–510), or LL5β(305–450). (**A**) Lysates (300 μg protein) were incubated with GFP-TRAP to immunoprecipitate full length GFP-LL5β. Filters with immunoprecipitates (IP) and lysates (30 μg) were blotted to detect endogenous ERC1 (filter **a**), GFP-LL5β (filter **b**), or the co-transfected LL5β fragments (filter **c**). (**B**) Quantification from n = 4 experiments as the one shown in (**A**) of the amount of endogenous ERC1 co-precipitating with GFP-LL5β. In each experiment, the amount of ERC1 coprecipitating with full length GFP-LL5β in the presence of βGalactosidase is considered as 100% of the interaction. One way ANOVA, Kruskal Wallis. Dunn's multiple comparisons. (**C**) The unbound fractions after immunoprecipitation with GFP-TRAP shown in (**A**) were used for a second round of immunoprecipitations with anti-FLAG Ab. Filters were blotted for endogenous ERC1 (filter **a**) that was strongly co-immunoprecipitated with FLAG-LL5β(381–510), but not with FLAG-LL5β(305–450); with anti-FLAG (filter **b**) to detect the immunoprecipitated fragments, which were virtually immunodepleted from the unbound fractions (30 μg protein/lane). The filter **a** was next incubated with anti-GFP Ab without prior stripping (**c**), to show that GFP-LL5β was absent from the immunoprecipitates of the LL5β fragments, and it was immunodepleted from the unbound fractions.

immunoprecipitates obtained by incubating the unbound fractions from the first immunopre-cipitations with anti-FLAG antibodies to pulldown the LL5β fragments revealed a clear signal for endogenous ERC1 coprecipitating with FLAG-LL5β(381–510), while no ERC1 signal was detected in control immunoprecipitates of FLAG-LL5β(305–450) or FLAG-β-galactosidase (Fig 5C). These findings indicate that the ERC1-binding fragment LL5β(381–510) interferes specifically with the interaction between the ERC1 and LL5β full length proteins.

## LL5 fragments are recruited as clients in ERC1 condensates

Endogenous LL5 proteins accumulate as clients in GFP-ERC1–positive condensates [13]. To test whether the ERC1 interacting LL5β fragments were recruited as client proteins at the

ERC1–induced condensates, we co-expressed either FLAG-LL5β(305–450) or FLAG-LL5β (381–510) with GFP-ERC1. The two Liprin-α1 constructs FLAG-Liprin-α1-EBR (the ERC1 binding region of Liprin-α1) and FLAG-Liprin-α1-ΔEBR (a deletion mutant missing the ERC1 binding region of Liprin-α1) were used as positive and negative controls for the specific localization at ERC1 condensates, respectively [13]. We estimated the colocalization of the FLAG-tagged LL5β constructs with the GFP-ERC1 condensates by Pearson's correlation coefficient [27]. As expected, FLAG-Liprin-α1-ΔEBR localized poorly at ERC1 condensates, while a clear colocalization of FLAG-Liprin-α1-EBR with ERC1 condensates was evident (S4A Fig). Intriguingly, both the ERC1-interacting LL5β(381–510) and LL5β(305–450) (not interacting with ERC1 by co-IP experiments) localized at ERC1 condensates (S4B Fig). Thus, the localization of the LL5β fragments at ERC1 condensates was independent from the capacity of the fragment to co-immunoprecipitate with ERC1. The possibility that the recruitment of the LL5β fragment at condensates is driven by dimerization/ oligomerization with endogenous LL5β is unlikely, since these fragments do not interact with full length LL5β (S1B Fig). On the other hand, this result suggests that the ability of ERC1–interacting fragment FLAG-LL5β (381–510) to interfere with the function of the complex formed by the full length proteins may be facilitated by the recruitment of the fragments as clients of the ERC1-induced condensates.

## The ERC1-interactig fragment LL5β(381–510) affects MDA-MB-231 tumor cell motility

We utilized a set of cell motility assays to determine the functional consequences of expressing the ERC1-binding fragment LL5β(381–510). The endogenous PMAP proteins ERC1 and LL5β are needed for efficient tumor cell migration [7]. We tested the effects of the ERC1-binding fragment LL5β(381–510) interfering with the formation of endogenous PMAPs and of LL5β/ ERC1 complexes, on 2D random migration by time-lapse experiments (5 h recording). Cells expressing GFP-LL5β(381–510) showed a non-significant mild reduction of 2D migration compared to control GFP expressing cells (S5A–S5C Fig). Human MDA-MB-231 cells are aggressive breast cancer cells with metastatic potential. To address the effect of LL5β(381–510) in an environment more related to the one met by invading tumor cells, we utilized a 3D migration assay in reconstituted extracellular matrices prepared from cultured NIH-3T3 fibroblast–like cells [28]. Both GFP-LL5β(305–450) and GFP-LL5β(381–510) reduced migration in the 3D environment compared to GFP transfected control cells (Fig 6A, 6B; S6 Fig). Cancer cells have the ability to degrade the extracellular matrix to invade host tissues. We used a gelatin degradation assay to measure the effects of the ERC1-interacting LL5β fragments on the potential of MDA-MB-231 cells to invade the extracellular matrix. Both spreading and extracellular matrix degradation were enhanced by Liprin-α1, as previously reported (S5D–S5F Fig). Spreading on extracellular matrix was not affected by LL5β(381–510) expression, and the decrease in extracellular matrix degradation observed in LL5β(381–510) expressing cells was not significant compared to control MDA-MB-231 cells (S5E Fig). Interestingly, this non-significant decrease in extracellular matrix degradation corresponded to a significant decrease in the density and number of invadopodia in cells expressing GFP-LL5β(381–510) compared to control cells expressing either βGalactosidase or GFP-LL5β(305–350) (Fig 6C–6E).

To test the effect of the expression of the ERC1 interacting fragment LL5β(381–510) on tumor cell invasion, we obtained lines of MDA-MB-231 cells stably expressing either LL5β (381–510), the non-interacting fragment LL5β(305–450), or GFP (Fig 7A and 7B). Displacement of endogenous ERC1 from PMAPs was confirmed in cells from clone expressing the ERC1-interacting fragment GFP-LL5β(381–510); in contrast, clones expressing either GFP or the non-interacting fragment GFP-LL5β(305–450) showed normal localization of endogenous

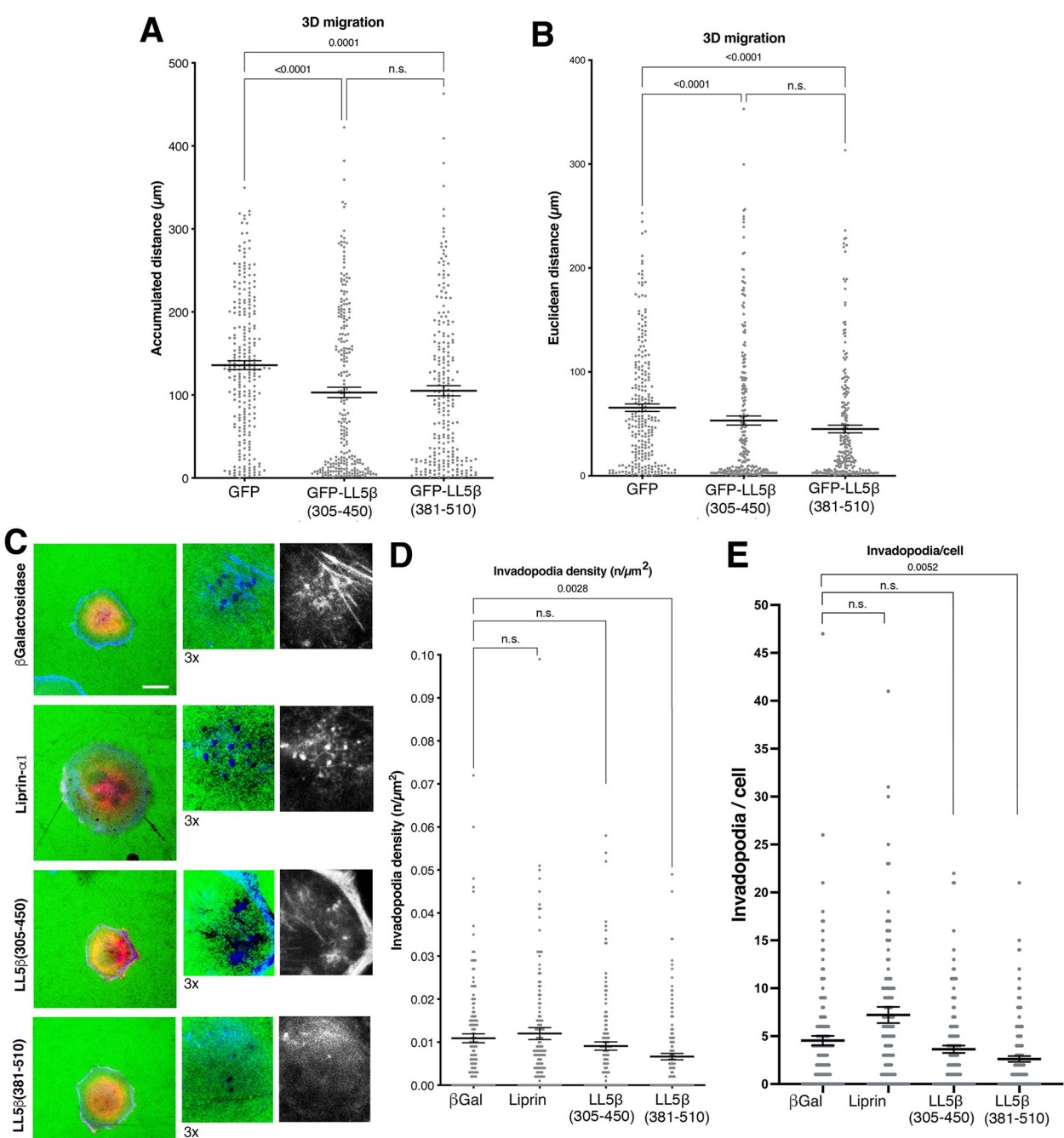

**Fig 6. GFP-LL5β(381–510) inhibits the migration of MDA-MB-231 cells in 3D extracellular matrix and the formation of invadopodia.** (**A-B**) Transfected MDA-MB-231 cells were plated in NIH-3T3-derived 3D extracellular matrix and their movement was tracked for 8 h. Graphs represent the accumulated distance (**A**) and the Euclidean distance (**B**). Mean ± SEM; n = 236–254 cells from 3 experiments. One-way ANOVA, Kruskal Wallis and Dunn's post hoc. (**C**) Cells plated on fluorescently labelled gelatin (same as in S5D Fig) were analyzed by immunofluorescence. F-actin staining was used to morphologically identify invadopodia (arrows to show examples in correspondence of black areas in the green channel corresponding to areas of extracellular matrix degradation). Bar, 20 μm. (**D-E**) Quantification of the density (**D**) and number of invadopodia per cell (**E**). Mean ±SEM; n = 136–146 cells from 3 experiments). One-way ANOVA, Kruskal Wallis and Dunn's post hoc.

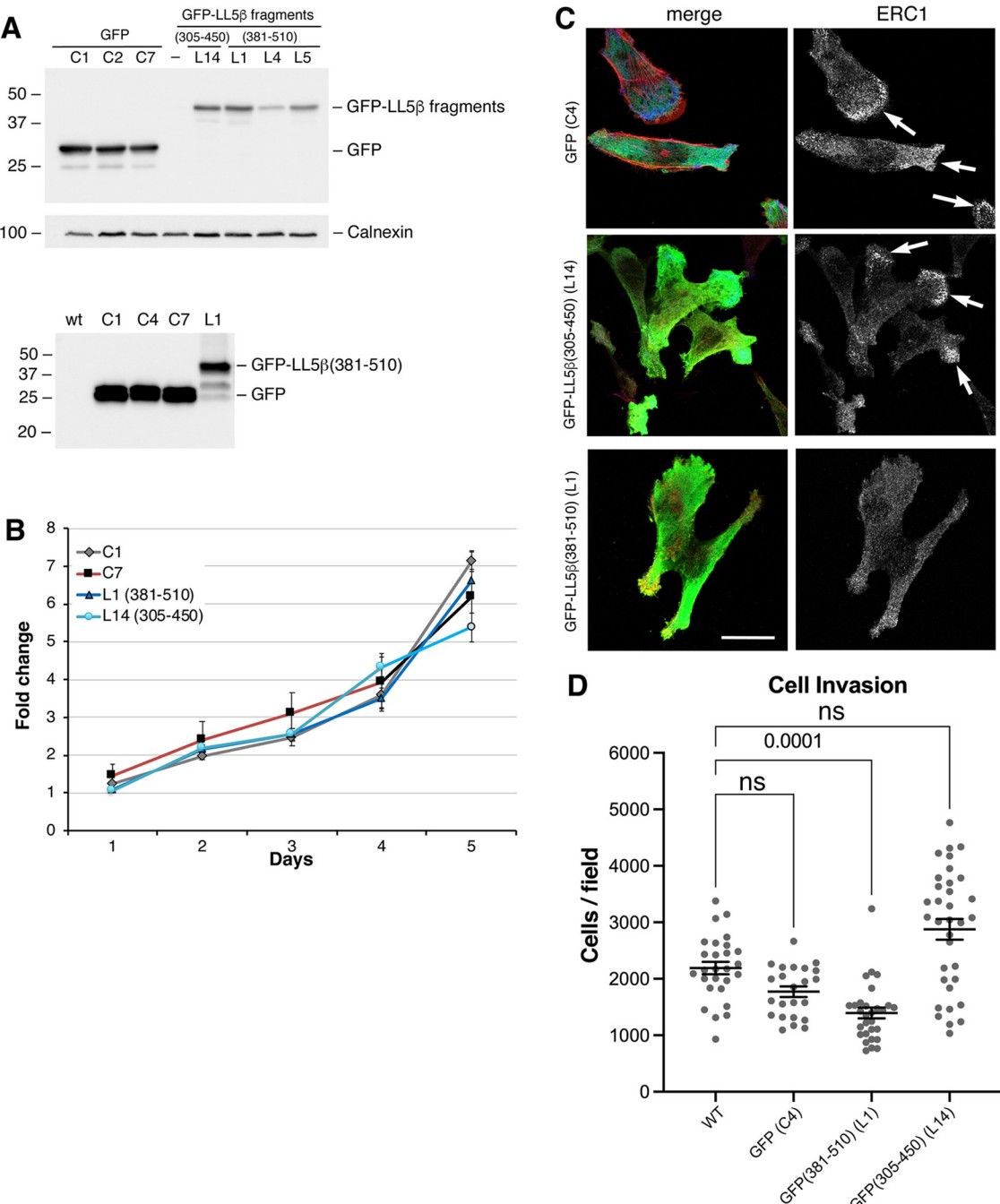

**Fig 7. GFP-LL5β(381–510) inhibits tumor cell invasion.** (**A**) Immunoblotting of clones from MDA-MB-231 cells transfected with the indicated GFP-tagged constructs (30 μg of protein lysate/lane). (**B**) MTT assays for clones derived from MDA-MB-231 cells transfected with the indicated GFP-tagged constructs. (**C**) Immunofluorescence shows dispersion of endogenous ERC1 from PMAPs in cell lines expressing GFP-LL5β(381–510) compared to cells expressing either GFP or control fragment GFP-LL5β(305–450). Arrows indicate examples of accumulation of endogenous ERC1 at peripheral PMAPs. Bar, 20 μm. (**D**) Clones of MDA-MB-231 cells expressing the indicated GFP-tagged proteins were used for transwell Matrigel invasion assays: invasion is significantly decreased for cells expressing GFP-LL5β(381–510), but not for cells expressing either GFP or GFP-LL5β(305–450). Mean ±SEM; n23 = 34 fields from 2–3 experiments. One-way ANOVA, Kruskal Wallis and Dunn's post hoc.

ERC1 accumulating near the edge of migrating cells (Fig 7C). Interestingly, GFP-LL5β(381–510) inhibited cell invasion determined by using Matrigel invasion assays, while no significant effects were detected by cells expressing either GFP or GFP-LL5β(305–450) control constructs (Fig 7D).

## Conclusions

Assembly of molecular networks at the leading edge is important to coordinate several processes needed for efficient protrusion during cell migration and invasion. These processes include cytoskeletal rearrangements, adhesion dynamics and extracellular matrix degradation. PMAPs include core scaffold proteins that are important for these processes [35]. In this direction, the PMAP core proteins ERC1/ELKS and LL5β are known to be required to facilitate the turnover of focal adhesions that is needed to promote cell advancement [22, 36]. In addition, LL5β is required to tether microtubules to adhesions via microtubule-associated CLASP proteins [37], and this mechanism has been proposed to establish a local secretory pathway to deliver metalloproteases that may allow both focal adhesion turnover and extracellular matrix degradation during invasion [36]. Our results support the hypothesis that it is possible to disrupt the assembly of PMAPs by interfering with the interaction between core components of the PMAPs. The observation that a LL5β fragment affecting the interaction between full length ERC1 and LL5β can inhibit processes required for tumor cell invasion suggests that the intermolecular interactions involved in the assembly of PMAPs in invasive tumor cells are interesting novel targets to be considered for anti-metastatic therapy.

Fusions of either LL5β or ERC1 with different genes including tyrosine kinase receptors (e.g. ERC1-RET) [33] have been identified in invasive breast cancer and other tumors (cBioPortal, TCGA). Moreover, tumor cells rely on several molecular pathways that are essential to their survival and function, such as invasion. ERC1/LL5 proteins are part of the PMAPs machinery that regulates tumor cell invasion. Understanding the biology and regulation of this protein network may be relevant not only when these proteins are altered in patients, but also for all tumors in which these proteins are necessary for invasion and the formation of metastases: an example of non-oncogene addiction genes [38], that are becoming attractive drug targets for combined cancer therapies [39].

## Supporting information

**S1 Raw images. Full uncropped blot images of panels presented in the Figures.** Unedited images of the blots and gels shown in the indicated panels of Figs 1, 2, 5, 7, and S1 Fig. The description of the experimental conditions are described in the legends of the respective Figures. (TIF)

**S1 Fig. LL5β and ERC1 fragments do not interact with full length LL5β and Liprin-α1, respectively.** (**A**) Aliquots of lysates (30 μg protein) from COS7 and MDA-MB-231 cells before (left filters) or after (right filters) immunoprecipitation of endogenous LL5α were blotted with the indicated specific antibodies to reveal endogenous LL5α, LL5β or ERC1. (**B**) Aliquots (250 μg protein) of lysates from COS7 cells co-transfected with the indicated constructs were used for immunoprecipitation with anti-GFP Abs. Membranes were cut and immunostained with the indicated Abs. (**C**) Aliquots (150 μg protein) of lysates from COS7 cells transfected with GFP–tagged ERC1 and FLAG-LL5β(381–510) constructs were immunoprecipitated with anti-FLAG Abs. Immunoprecipitates (IP) and lysates (30 μg) were immunoblotted to detect the tagged constructs and endogenous Liprin-α1. (TIF)

**S2 Fig. Expression of ERC1(270–370) fragments in migrating MDA-MB-231 cells.** Localization of endogenous Liprin-α1 (**A**) and LL5 proteins (**B**) in migrating MDA-MB-231 cells transfected with the indicated constructs. Bars, 20 μm.
(TIF)

**S3 Fig. Expression of LL5β(381–510) fragments in migrating MDA-MB-231 cells.** Localization of endogenous Liprin-α1 (**A**) and LL5 proteins (**B**) in migrating MDA-MB-231 cells transfected with the indicated constructs. Bars, 20 μm.
(TIF)

**S4 Fig. LL5β fragments are recruited at GFP-ERC1 condensates.** (**A**) Specific recruitment of FLAG-tagged (blue) LL5β fragments at GFP-ERC1 (green) condensates in co-transfected COS7 cells. Bar = 20 μm. Three-fold enlargements of areas indicated by arrows. (**B**) Pearson's correlation coefficient for the colocalization of fragments at ERC1 condensates. Mean ±SEM; n = 54–89 condensates analyzed for each condition from 3 experiments. ANOVA, Tukey post hoc; FLAG-Liprin-α1-ΔEBR as negative control.
(TIF)

**S5 Fig. Effects of LL5β fragments on cell motility and extracellular matrix degradation.** (**A-B**) 2D random cell migration: accumulated distance (**A**) and Euclidean distance (**B**) of transfected MDA-MB-231 cells plated on 2.5 μg/ml FN and tracked for 5 h. Mean ±SEM; n = 194–208 cells from 3 experiments. One way ANOVA, Kruskal-Wallis test, Dunn's multiple comparisons. (**C**) Trajectories of MDA-MB-231 cells transfected with the indicated constructs, plated in NIH-3T3-derived 3D extracellular matrix, and tracked for 8 h (n = 81–84 cells per experimental condition). (**D-F**) LL5β(381–510) does not affect extracellular matrix degradation. Transfected MDA-MB-231 cells were plated on fluorescently labelled gelatin. After 5 hours the dark areas of gelatin degradation (**D**; bar, 20 μm) were quantified (**E**). For each cell analyzed, the gelatin degradation area was normalized to the corresponding projected cell area (**F**). Mean ±SEM; n = 186–194 cells from 4 experiments. One way ANOVA, Kruskal-Wallis test, Dunn's multiple comparisons.
(TIF)

**S6 Fig. 3D random cell migration.** Trajectories of MDA-MB-231 cells transfected with the indicated constructs, plated in NIH-3T3-derived 3D extracellular matrix, and tracked for 8 h (n = 81–84 cells per experimental condition).
(TIF)

## Acknowledgments

We thank the personnel of the Advanced Light and Electron Microscopy BioImaging Center (ALEMBIC) of the San Raffaele Scientific Institute for technical support, Prof. Isabella Felli (University of Florence, Italy) for helpful discussion, and Dr. Yuko Mimori-Kiyosue (RIKEN Center for Biosystems Dynamics Research, Kobe, Japan) for the LL5α antibody. This work benefited from access to the Protein Production in *E. coli* with Isotope Labelling for NMR platform at CERM/CIRMMP (Florence, Italy), an Instruct-ERIC centre.

## Author Contributions

**Conceptualization:** Lucrezia Maria Ribolla, Kristyna Sala, Roberta Pierattelli, Alessandro Provenzani, Ivan de Curtis.

**Data curation:** Lucrezia Maria Ribolla, Kristyna Sala, Diletta Tonoli, Martina Ramella, Lorenzo Bracaglia, Ivan de Curtis.

**Formal analysis:** Roberta Pierattelli, Ivan de Curtis.

**Funding acquisition:** Ivan de Curtis.

**Investigation:** Lucrezia Maria Ribolla, Kristyna Sala, Martina Ramella, Lorenzo Bracaglia, Isabelle Bonomo, Leonardo Gonnelli, Andrea Lamarca, Matteo Brindisi, Alessandro Provenzani.

**Methodology:** Lucrezia Maria Ribolla, Kristyna Sala, Diletta Tonoli, Martina Ramella, Lorenzo Bracaglia, Isabelle Bonomo, Leonardo Gonnelli, Andrea Lamarca, Matteo Brindisi, Roberta Pierattelli, Alessandro Provenzani.

**Project administration:** Ivan de Curtis.

**Resources:** Ivan de Curtis.

**Supervision:** Leonardo Gonnelli, Roberta Pierattelli, Alessandro Provenzani, Ivan de Curtis.

**Validation:** Lucrezia Maria Ribolla, Kristyna Sala, Diletta Tonoli, Ivan de Curtis.

**Writing – original draft:** Ivan de Curtis.

**Writing – review & editing:** Lucrezia Maria Ribolla, Kristyna Sala, Lorenzo Bracaglia, Roberta Pierattelli, Alessandro Provenzani, Ivan de Curtis.

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
