## [Decision Letter · Decision Letter 0]

22 May 2023

PONE-D-23-13577Interfering with the ERC1–LL5β interaction disrupts plasma membrane–associated platforms and affects tumor cell motilityPLOS ONE

Dear Dr. de Curtis,

Thank you for submitting your manuscript to PLOS ONE. After careful consideration, we feel that it has merit but does not fully meet PLOS ONE’s publication criteria as it currently stands. Therefore, we invite you to submit a revised version of the manuscript that addresses the points raised during the review process.

We look forward to receiving your revised manuscript.

Kind regards,

Zhiming Li, Ph.D.

Academic Editor

PLOS ONE

Journal Requirements:

"We thank the personnel of the Advanced Light and Electron Microscopy BioImaging Center (ALEMBIC) of the San Raffaele Scientific Institute for technical support, Prof. Isabella Felli (University of Florence, Italy) for helpful discussion,  and Dr. Yuko Mimori-Kiyosue (RIKEN Center for Biosystems Dynamics Research, Kobe, Japan) for the LL5� antibody. IdC was supported by a grant from AIRC—Associazione Italiana per la Ricerca sul Cancro (IG 2017 Id.20203). KS and MR were supported by postdoctoral fellowships from AIRC. LMR was supported by a FCSR-Fronzaroli fellowship; work performed by LMR was in partial fulfilment of the requirements for obtaining the PhD degree at the Vita-Salute San Raffaele University, Milano, Italy. This work benefited from access to the Protein Production in E. coli with Isotope Labelling for NMR platform at CERM/CIRMMP (Florence, Italy), an Instruct-ERIC centre. Financial support was provided by Instruct-ERIC (PID: 17416). "

"IdC was supported by AIRC—Associazione Italiana per la Ricerca sul Cancro (IG 2017 Id.20203), and by Instruct-ERIC (PID: 17416).

KS and MR were supported by postdoctoral fellowships from AIRC.

LMR was supported by a FCSR-Fronzaroli fellowship; work performed by LMR was in partial fulfilment of the requirements for obtaining the PhD degree at the Vita-Salute San Raffaele University, Milano, Italy.

Additional Editor Comments:

Dear Dr. Ivan de Curtis,

Thank you for submitting your manuscript to PLOS One. Your work has now been evaluated by two referees and both of them showed great interest in the story. There are, however, a few concerns that need to be addressed before we can move forward. I hope that you find these comments helpful to improve the manuscript.

Reviewers' comments:

Reviewer's Responses to Questions

**Comments to the Author**

1. Is the manuscript technically sound, and do the data support the conclusions?

Reviewer #1: Yes

Reviewer #2: Partly

2. Has the statistical analysis been performed appropriately and rigorously? 

Reviewer #1: Yes

Reviewer #2: Yes

3. Have the authors made all data underlying the findings in their manuscript fully available?

Reviewer #1: Yes

Reviewer #2: Yes

4. Is the manuscript presented in an intelligible fashion and written in standard English?

Reviewer #1: Yes

Reviewer #2: Yes

5. Review Comments to the Author

Reviewer #1: The submitted manuscript written by Ivan de Curtis et al. analyses extensively the interacting sites of the ERC1–LL5β binding. The manuscript is overall well and clearly written, I really liked the proposed idea that LL5β fragment inhibiting the interaction between ERC1 and LL5β leads to the prevention of the complex formation of these adaptor proteins and interferes with the processes required for tumour cell invasion.

I have some suggestions for improving the manuscript, as in this form has some insufficiency:

1., Several interesting topics come up during the results part which are not mentioned earlier and introduced properly. Please insert some sentences in the introduction about these topics:

-Differences and similarities of LL5β and LL5alfa regarding their effects, functions, and structure.

-How LL5β serve as a bridge between the actin cytoskeletal system and the microtubule system in cells and regulates via partner molecules both filament nets?

-Molecular structure (domains) of ERC1.

-Insert a sentence describing the differences between lamellipodia and invadopodia.

2., I would include an overview about the already published results (or search in databases) about the expression level changes of ERC1 and LL5β in cancers (especially it would be most relevant in breast cancer, as the MDA-MB-231 is a human breast adenocarcinoma cell line used in this study). This would fit at the very end of the results part.

3., Need reference for this sentence: Line 364: “In COS7 cells the endogenous LL5 protein is represented mostly by LL5alfa”

4., I would suggest to perform a PLA assay (Duolink) in a cell line expressing endogenous LL5β and endogenous ERC1. In this setting, it would be very convincing to show that the introduction of LL5β(381-510) disrupts the PLA signal (for endogenous full-LL5β and endogenous ERC1). Because the manuscript states that: “line 580: These findings indicate that the ERC1-binding fragment LL5β(381-510) interferes specifically with the interaction between the ERC1 and LL5 full length proteins.” Actually, the transduced LL5β full length protein was measured not the endogenous full length LL5β proteins in the cancer cells.

5., Anyway, it is not clear if the MDA-MB-231 cell express or not the endogenous LL5β full length protein?

Reviewer #2: This manuscript deals with ERC1-LL5beta interaction in breast cancer cells. In particular, the binding sites of these molecules and the inhibitory effects of LL5beta fragments were researched well. However, some data and description are missing

Major points:

1. In the breast cancer motility, the author showed ERC1 accumulation in PMAP. And,

the fragment of ERC1 inhibits the accumulation of ERC1. The author mentioned that the inhibition of accumulating ERC1 in PMAP decreases the cell motility. However, the molecular function of ERC1 in cell migration is not mentioned. The author should mention the mechanism and show the morphological change of the cell and decreasing the any other PMAP components by adding the fragments.

2. The formation of invadopodia is multistep process. The function of ERC1 in forming the invadopodia is not mentioned. Which molecules do ERC1 participate in the formation of invasive protrusions?　How signal is the pathway induced by ERC1? The author needs to describe. In addition, the author should perform the experiments showing the invadopodial signal transduction including ERC1 and inactivating by adding fragments.

3. In quantitative evaluation of invadopoaia, the author used invadopodial number par

area. As far as I know, usually, the gelatin degeneration area per cell is correlate the invasive ability. Why the author used the unit “ invadopodial number per” ?

4. If full length-LL5beta can make dimer/oligomer and the fragment can not, the GFP (with fulllength/fragment LL5beta) density of the ERC1 accumulating legion may be different. Therefore, the correlation of GFP and ERC1 intensity should be measured and compared.

Minor points:

1. In the figure of immunoblotting, the author described the weight of samples.

Is it exact, and important?

2. In the experiments of cell migration, the typical trajectories of control and should be shown.

3. The reference to GFP (305-450) and GFP(381-510) in Figure 6 is misleading and should be corrected to LL5β-GFP ().

6. PLOS authors have the option to publish the peer review history of their article (what does this mean?). If published, this will include your full peer review and any attached files.

Reviewer #1: No

Reviewer #2: No

---

## [Author Response · Author response to Decision Letter 0]

30 May 2023

Editor Comments:

Thank you for submitting your manuscript to PLOS One. Your work has now been evaluated by two referees and both of them showed great interest in the story. There are, however, a few concerns that need to be addressed before we can move forward. I hope that you find these comments helpful to improve the manuscript.

We thank the Editor and the two Reviewers for the useful suggestions that have helped us to improve our study. Following are our answers to the points raised by the two Reviewers (reported below in italics). In the file “Revised Manuscript with Track Changes” changes have been underlined and highlighted in yellow.

Reviewer's Responses to Questions

1. Is the manuscript technically sound, and do the data support the conclusions?

Reviewer #1: Yes

Reviewer #2: Partly

We hope that following the revision, the manuscript will now be acceptable for publication in PLOS ONE.

Reviewer #1:

1. Several interesting topics come up during the results part which are not mentioned earlier and introduced properly. Please insert some sentences in the introduction about these topics:

-Differences and similarities of LL5β and LL5alfa regarding their effects, functions, and structure.

-How LL5β serve as a bridge between the actin cytoskeletal system and the microtubule system in cells and regulates via partner molecules both filament nets?

- Molecular structure (domains) of ERC1.

- Insert a sentence describing the differences between lamellipodia and invadopodia.

We have now introduced the information in the Introduction, at pages 4, 5, and 6.

2. I would include an overview about the already published results (or search in databases) about the expression level changes of ERC1 and LL5β in cancers (especially it would be most relevant in breast cancer, as the MDA-MB-231 is a human breast adenocarcinoma cell line used in this study). This would fit at the very end of the results part.

This is a nice suggestion allowing us to better explain the importance of studying these proteins with respect to cancer. In fact, fusions of either LL5� or ERC1 with different genes including tyrosine kinase receptors (e.g. ERC1-RET) [Nakata et al. 1999] have been identified in invasive breast cancer and other tumors (cBioPortal, TCGA). Moreover, tumor cells rely on several molecular pathways that are essential to their survival and function, such as invasion. ERC1/LL5 proteins are part of the PMAPs machinery that regulates tumor cell invasion. Understanding the biology and regulation of this protein network may be relevant not only when these proteins are altered in patients, but also for all tumors in which these proteins are necessary for invasion and the formation of metastases: an example of non-oncogene addiction genes [Luo et al, 2009], that are becoming attractive drug targets for combined cancer therapies [Petroni et al, 2022]. This part has been added to the conclusions, at page 31.

3. Need reference for this sentence: Line 364: “In COS7 cells the endogenous LL5 protein is represented mostly by LL5alfa”

Since no data have been published yet on the relative expression of LL5� and � in COS7 cells, we have now added a new panel S1A Fig, and modified the sequence of panels of S1 Fig in the text accordingly. The new panel indicates relative lower expression of LL5� in COS7 with respect to LL5�, and relative to LL5� expression in MDA-MB-231 cells.

4. I would suggest to perform a PLA assay (Duolink) in a cell line expressing endogenous LL5β and endogenous ERC1. In this setting, it would be very convincing to show that the introduction of LL5β(381-510) disrupts the PLA signal (for endogenous full-LL5β and endogenous ERC1). Because the manuscript states that: “line 580: These findings indicate that the ERC1-binding fragment LL5β(381-510) interferes specifically with the interaction between the ERC1 and LL5 full length proteins.” Actually, the transduced LL5β full length protein was measured not the endogenous full length LL5β proteins in the cancer cells.

The approach suggested by the Reviewer is interesting. On the other hand it represents an addition requiring quite an extensive effort for this lab, since we have to set conditions for a technique not in use in this lab. We will consider this approach for a future extension of the present work. 

5. Anyway, it is not clear if the MDA-MB-231 cell express or not the endogenous LL5β full length protein?

The expression of the endogenous LL5� protein in MDA-MB-231 cells was carefully shown by using isoform-specific antibodies in our previous publication: Astro et al. 2014 Liprin-α1, ERC1 and LL5 define polarized and dynamic structures that are implicated in cell migration. J Cell Sci 127:3862 (see Figure 1 of this publication). 

We have now made this clear by mentioning this reference at page 17.

Reviewer #2:

1. In the breast cancer motility, the author showed ERC1 accumulation in PMAP. And, the fragment of ERC1 inhibits the accumulation of ERC1. The author mentioned that the inhibition of accumulating ERC1 in PMAP decreases the cell motility. However, the molecular function of ERC1 in cell migration is not mentioned. The author should mention the mechanism and show the morphological change of the cell and decreasing the any other PMAP components by adding the fragments.

The exact molecular function of ERC1 in cell migration is not known yet. We hypothesize that ERC1 is needed to assemble the PMAPs that promote protrusion and the turnover of focal adhesions, as previously shown (Astro et al, 2016). We had mentioned this in the conclusions. Of note, while the fragments affect the localization of endogenous ERC1 at PMAPs, we found that they did not affect the localization of the other endogenous PMAP components liprin-�1 and LL5: this is shown in supplementary figures S2 Fig and S3 Fig. 

2. The formation of invadopodia is multistep process. The function of ERC1 in forming the invadopodia is not mentioned. Which molecules do ERC1 participate in the formation of invasive protrusions?　How signal is the pathway induced by ERC1? The author needs to describe. In addition, the author should perform the experiments showing the invadopodial signal transduction including ERC1 and inactivating by adding fragments.

The sequence of molecular events linking ERC1 to the formation of invadopodia is still unknown: this topic is interesting but goes beyond the scope of this manuscript. 

3. In quantitative evaluation of invadopoaia, the author used invadopodial number par

area. As far as I know, usually, the gelatin degeneration area per cell is correlate the invasive ability. Why the author used the unit “ invadopodial number per” ?

We have used the density of invadopodia to correct for the different projected cell areas. We have now added also the quantification of the total number of invadopodia per cell (new Fig 6E).

4. If full length-LL5beta can make dimer/oligomer and the fragment can not, the GFP (with fulllength/fragment LL5beta) density of the ERC1 accumulating legion may be different. Therefore, the correlation of GFP and ERC1 intensity should be measured and compared.

In Figure 4, while full length LL5� accumulates at the ERC1-positive PMAPs, the fragment LL5�(381-510) is diffuse; in addition this fragment induces the almost complete redistribution of ERC1 from PMAPs. This result suggest that the LL5beta fragment, but not the full length LL5beta, displaces endogenous ERC1 from PMAPs. Of note, the LL5beta fragment does not displace the endogenous LL5beta and Liprin-alpha1 (S3 Fig). 

Minor points:

1. In the figure of immunoblotting, the author described the weight of samples.

Is it exact, and important?

The numbers on the side of the blots in Figs 1 and 2 indicated the amount of protein loaded per lane. We have now removed the information.

2. In the experiments of cell migration, the typical trajectories of control and should be shown.

Trajectories have been added in S5 Fig (new panel C, for a typical 2D experiment) and in the new S6 Fig (for a typical 3D experiment).

3. The reference to GFP (305-450) and GFP(381-510) in Figure 6 is misleading and should be corrected to LL5β-GFP ().

Correction made.

---

## [Decision Letter · Decision Letter 1]

12 Jun 2023

Interfering with the ERC1–LL5b interaction disrupts plasma membrane–associated platforms and affects tumor cell motility

PONE-D-23-13577R1

Dear Dr. de Curtis,

We’re pleased to inform you that your manuscript has been judged scientifically suitable for publication and will be formally accepted for publication once it meets all outstanding technical requirements.

Kind regards,

Zhiming Li, Ph.D.

Academic Editor

PLOS ONE

Reviewers' comments:

Reviewer's Responses to Questions

**Comments to the Author**

1. If the authors have adequately addressed your comments raised in a previous round of review and you feel that this manuscript is now acceptable for publication, you may indicate that here to bypass the “Comments to the Author” section, enter your conflict of interest statement in the “Confidential to Editor” section, and submit your "Accept" recommendation.

Reviewer #1: All comments have been addressed

Reviewer #2: (No Response)

2. Is the manuscript technically sound, and do the data support the conclusions?

Reviewer #1: Yes

Reviewer #2: Yes

3. Has the statistical analysis been performed appropriately and rigorously? 

Reviewer #1: Yes

Reviewer #2: Yes

4. Have the authors made all data underlying the findings in their manuscript fully available?

Reviewer #1: Yes

Reviewer #2: Yes

5. Is the manuscript presented in an intelligible fashion and written in standard English?

Reviewer #1: Yes

Reviewer #2: Yes

6. Review Comments to the Author

Reviewer #1: All points raised during the review process were addressed and all the corrections were made. This manuscript is suitable for accepting in PLOS ONE and the publishing process can move forward.

Reviewer #2: I apologize for the delay in the review of this revised manuscript. The author was already addressed my question and proposal.

7. PLOS authors have the option to publish the peer review history of their article (what does this mean?). If published, this will include your full peer review and any attached files.

Reviewer #1: No

Reviewer #2: No

---

## [Editor Report · Acceptance letter]

4 Jul 2023

PONE-D-23-13577R1 

Interfering with the ERC1–LL5b interaction disrupts plasma membrane–associated platforms and affects tumor cell motility 

Dear Dr. de Curtis:

I'm pleased to inform you that your manuscript has been deemed suitable for publication in PLOS ONE. Congratulations! Your manuscript is now with our production department. 

Kind regards, 

on behalf of

Dr. Zhiming Li 

Academic Editor

PLOS ONE